# Circular RNAs in Cardiovascular Physiopathology: From Molecular Mechanisms to Therapeutic Opportunities

**DOI:** 10.3390/ijms26199725

**Published:** 2025-10-06

**Authors:** Giorgia Capirossi, Sofia Brasini, Elena Tremoli, Andrea Binatti, Roberta Roncarati

**Affiliations:** 1Maria Cecilia Hospital, GVM Care & Research, 48033 Cotignola, Italy; gcapirossi@gvmnet.it (G.C.); sbrasini@gvmnet.it (S.B.); etremoli@gvmnet.it (E.T.); rroncarati@gvmnet.it (R.R.); 2CNR Institute of Molecular Genetics “Luigi Luca Cavalli-Sforza”, Unit of Bologna, 40136 Bologna, Italy

**Keywords:** circular RNAs (circRNAs), cardiovascular diseases (CVDs), biogenesis, methodologies, detection, validation, translational medicine, biomarkers, RNA therapy

## Abstract

Circular RNAs are a class of stable non-coding RNAs generated through a back-splicing mechanism. They are now recognized as central players in cell function and are no longer considered byproducts of transcription. CircRNAs regulate gene expression at the transcriptional, post-transcriptional, and translational levels by interacting with various molecules. They act as sponges for miRNAs and proteins, molecular scaffolds, and can also be translated into peptides. Although advances in next-generation sequencing and PCR methods have improved their identification and quantification, technical and bioinformatic challenges remain. Increasing evidence shows their involvement in cardiovascular diseases such as heart failure, hypertrophy, fibrosis, and atherosclerosis, with protective or deleterious effects depending on the context. Given their presence in biological fluids and extracellular vesicles, they can be considered promising biomarkers, but their therapeutic applications are still under investigation. Future studies including a better understanding of their mechanisms of action, the development of standardized validation methods, and potential clinical applications (prevention, early diagnosis, personalized therapies) in diseases are still needed. This review provides an updated overview of the knowledge regarding circRNAs and their translational role in health and disease with a particular focus on cardiovascular diseases.

## 1. Introduction

More than two decades after the initial publication of the human genome, scientists have sequenced its 3 billion DNA bases and identified thousands of genes. Yet, the complete meaning and interpretation of our DNA remains incomplete. Of the whole human genome, only 2% of genes encode proteins. Most of our genetic material is classified as non-coding DNA [1], which plays a critical role in regulating transcription and translation processes [2]. Among the different classes of non-coding RNAs, circular RNAs (circRNAs) have emerged as important key regulators of gene expression and the transcription process by acting through a competitive microRNA-binding (miRNA) mechanism [3]. These molecules were initially discovered in 1976, in plant viroids [4], and their existence in eukaryotic cells was subsequently demonstrated in 1979 by Hsu and Coca-Prados using electron microscopy in HeLa cells [5]. CircRNAs are single-stranded RNA molecules generated through a back-splicing mechanism, characterized by the covalent linking of their 5′ and 3′ ends to form a closed loop structure. The absence of free ends gives circRNAs greater stability and resistance to exonuclease than linear RNAs, making them highly stable in cellular environments [6]. Thanks to their mechanism of action in regulating transcription, circRNAs have been increasingly implicated in the pathogenesis of various diseases, including cardiovascular disorders [7]. In addition, their intrinsic stability makes them promising biomarkers for various pathological conditions [8]. In this review we will discuss the main characteristics, biological functions, and detection techniques of circRNAs, emphasizing their emerging translational potential in cardiovascular disease (CVDs).

## 2. Biogenesis CircRNAs

Given that the majority of circRNAs consist of exons located within the internal segments of the coding sequence, their formation is primarily thought to be linked to RNA splicing mechanisms [9]. The canonical splicing pathway is the main process through which spliceosomes remove introns from pre-mRNAs and join exons to produce linear mRNA transcripts. However, in addition to generating linear mRNAs, pre-mRNAs can also act as precursors to circRNAs via a distinct noncanonical splicing process known as back-splicing. Despite advances in understanding, the precise regulatory mechanisms controlling circRNA biogenesis remain partially unclear. Functional studies in animal models and human cells have shed light on the interplay between canonical splicing and circRNA formation, although some findings present contradictions. For instance, research on Drosophila melanogaster has demonstrated that inhibition of the spliceosome—by depleting U2 snRNP components—increases the ratio of circRNA to linear RNA transcripts, suggesting that delayed or impaired canonical splicing favors back-splicing and circRNA formation [10]. In contrast, studies conducted on human cell lines (e.g., HeLa cells) using the splicing inhibitor isoginkgetin, have shown a reduction in circRNA synthesis upon spliceosome assembly inhibition, indicating that circRNA biogenesis largely depends on spliceosome activity and canonical splicing machinery. These data confirm that the majority of circRNAs are generated co-transcriptionally through a spliceosome-dependent process called back-splicing [11,12]. In this process, the 5′ splice donor-site covalently links to an upstream 3′ splice acceptor-site, within the same pre-mRNA molecule, resulting in a covalently closed, stable circRNA, lacking the conventional 5′ cap and 3′ polyadenylated tail [13] (Figure 1).

Because circRNAs and their linear counterparts originate from the same pre-mRNA, the distinctive feature that allows specific identification of circRNAs is the back-splice junction (BSJ) that remains the unique sequence feature formed by the noncanonical splicing event [9]. Specifically, the circularization of intronic sequences is formed by the process that brings the downstream 5′ splice donor site and the upstream 3′ splice acceptor site into proximity. At the molecular level, circularization is thought to be facilitated by bringing the downstream 5′ splice donor site and the upstream 3′ splice acceptor site into close proximity. This spatial arrangement can be mediated by base pairing flanking inverted repeat elements such as Alu sequences, present upstream and downstream of the circularized exon(s) [14], or by dimerization of RNA-binding proteins (RBPs) that bind specific sequences in neighboring introns and promote back-splicing [15]. These mechanisms are illustrated in Figure 1c.

Additionally, during exon-skipping events, circRNAs may derive from lariat structures when internal back-splicing occurs on the lariat, creating circular products [16,17] (Figure 1a). Notably, circRNAs can also originate from intronic lariats generated by canonical splicing, which normally undergo debranching (hydrolysis of the 2′–5′ phosphodiester bond) and degradation, but may escape this process due to specific sequence motifs near the junction, such as GU-rich or C-rich elements [17]. This alternative origin of circRNAs is also depicted in Figure 1b.

### 2.1. Regulation of CircRNA Biogenesis

circRNA biogenesis can be influenced by gene regulatory elements. Many circRNAs are typically generated through the back-splicing of exons flanked by adjacent long introns in genes with highly active promoters [18,19,20]. Furthermore, epigenetic modifications of histones and genes’ regulatory regions influence alternative splicing and may directly impact circRNA biogenesis [21,22].

Research on *Drosophila melanogaster* suggests that biogenesis of circRNAs is also influenced by cis-acting and trans-acting splicing factors, including heterogeneous nuclear ribonucleoproteins (hnRNPs) and serine-arginine-rich (SR) proteins, indicating a complex interplay of factors that regulate where and when circRNAs are formed [23].

Two distinct classes of enzymes regulate circRNA biogenesis: (1) a class that reduces circRNA formation, which includes adenosine deaminase acting on RNA 1 (ADAR1) and the ATP-dependent RNA helicase A (DHX9); (2) a class that promotes circRNA production, consisting of NF90 and NF110, two protein isoforms encoded by the interleukin enhancer-binding factor 3 (ILF3) gene. NF90 and NF110 are involved in antiviral responses and enhance circRNA biogenesis [24]. Thus, the role of these factors can be clearly distinguished: ADAR1 and DHX9 inhibit circRNA formation by unwinding RNA (dsRNA) (Figure 1e), whereas NF90 and NF110 facilitate circRNA production by stabilizing dsRNA structures [25] (Figure 1d).

### 2.2. CircRNA Classification

CircRNAs can be classified based on their sequence composition and location (e.g., exons, introns, exon-intron) as well as their chromosomal origin, including gene-derived, intergenic, read-through transcripts or those arising from rearrangements). Following these features, circRNAs are commonly categorized into five major classes [26,27].

Exon circRNAs (ecircRNAs) represent the most abundant type and are formed exclusively through the back-splicing of exonic regions. Their sequence composition corresponds to that of their linear mRNA counterpart, although they lack the 5′ cap and 3′ poly-adenylated tail due to their covalently closed circular structure [28]. This process is often facilitated by complementary repetitive elements in flanking introns (e.g., Alu elements in mammals) or by the dimerization of RNA-binding proteins that bring splicing sites into proximity. EcircRNAs are predominately localized in the cytoplasm, present resistance to the RNase R-digestion, and are frequently involved in post-transcriptional regulatory functions (miRNA sponges, protein scaffolds, and, in some cases, translated into peptides [29]).Intronic circRNAs (ciRNAs) originate exclusively from intronic sequences and are generally less abundant and less stable than ecircRNAs. Their formation and stability rely on specific sequence motifs located near the branch point, such as a GU-rich element close to the 5′ site and a C-rich region near the branch point, which inhibit enzymatic debranching and facilitate the maintenance of the circular structure. ciRNAs predominantly accumulate in the nucleus, where they play a role in cis regulation of transcription [15].Exon-intron circRNAs (EIciRNAs) contain both exonic and intronic sequences [30]. These circRNAs are primarily localized in the nucleus where they interact with U1 small nuclear ribonucleoprotein (snRNP) and RNA polymerase II to promote cis regulatory transcription of their parental gene, participating in a transcriptional feedback mechanism [31].Read-through circRNAs (rt-circRNAs) result from the union of exons from two adjacent genes, which are typically oriented in the same transcriptional direction. This process, called transcription read-through, occurs when the RNA polymerase II enzyme fails to stop at the normal termination site of a gene and continues transcribing into the next downstream gene. Their resulting elongated RNA, which contains exons from the previous and subsequent genes without the intergenic region, is often more complex and difficult to annotate bioinformatically, because it contains two distinct genes. Their existence highlights the flexibility of RNA processing mechanisms and may have practical or diagnostic significance, as some rt-circRNAs are associated with tissue-specific expression patterns or relation with pathological conditions. These molecules are examples of how cells can use non-canonical splicing events to generate transcripts with potentially novel functions [32,33].Fusion-circRNAs (f-circRNAs) originate from chromosomal rearrangements such as translocations, inversions, and deletions, which are commonly observed in pathological conditions. They are generated by back-splicing of fusion gene transcripts, whereas exons, introns, or both from two non-adjacent genes are transcribed together. The back-splicing of these chimeric transcripts results in the formation of f-circRNAs, which are detectable only in cells harboring specific genomic rearrangements. Oncological studies have demonstrated that such rearrangements can give rise to f-circRNAs with distinct biological functions as represented by f-circM9 [34] whose tumorigenic properties have been experimentally tested in cellular and animal models [26].

To complete the circRNA classification it is essential to include their biogenesis mechanisms, as previously stated, and functional diversity, which will be described in detail later in the review.

## 3. Functions

CircRNA expression has been observed across species, ranging from prokaryotes to complex eukaryotes, indicating their fundamental biological role [35]. Although the BSJs of orthologous circRNAs in mammals often present low primary sequence conservation, their functions are likely preserved over evolutionary time and may depend on alternative features, like RNA secondary structure, rather than on the primary sequence alone [36]. Their widespread conserved function among different species supports the use of circRNAs in physiopathological research models. This is particularly relevant in areas such as CVDs, where animal models remain indispensable for investigating disease mechanisms and potential therapeutic interventions. Based on their sequence composition (as described in the previous paragraph), circRNAs are present in different subcellular compartments, where they have different functions. CircRNAs are predominantly localized in the cytoplasm, where many act as regulators of post-transcriptional gene expression. However, a smaller fraction resides in the nucleus, where they participate in regulating transcriptional processes [37]. Functionally, many circRNAs influence gene regulation at both transcriptional and post-transcriptional levels, while only a few have been shown to play a role in the translation process [15] (Figure 2). Their various functions have important implications for a wide range of diseases, including CVDs, cancer, and neurological disorders, highlighting their critical roles in both cellular physiology and pathology.

### 3.1. miRNA Sponging

Initially, the most studied functions of circRNAs were their role as miRNA sponges. CircRNAs in fact harbor microRNA response elements (MREs), a specific sequence that complementarily binds miRNAs and acts as a sponge, preventing miRNAs from interacting with their target mRNAs, and thus modulating gene expression [29,38]. These interactions are regulated by a precise balance, or stoichiometric relationship [39]. Since many circRNAs contain multiple miRNA binding sites, they can act as sponges to sequester several miRNAs in the cytoplasm, reducing their interaction with potential target mRNAs and consequently leading to an upregulation of the miRNA-regulated genes [15,40]. The first circRNA characterized in this scenario was CDR1as (ciRS-7), which contains over 70 conserved binding sites for miRNAs, with a particularly high affinity for miR-7 [41]. Since this discovery, numerous other circRNAs functioning as competitive endogenous RNAs (ceRNAs) have been identified, including circHIPK3 [42]. More recently, CircNCX1 has been demonstrated to function as a sponge for miR-133a-3p, preventing miR-133a-3p from repressing *CDIP1* mRNA during myocardial ischemia–reperfusion (MI/R) injury, which in turn promotes cardiomyocyte apoptosis [43].

### 3.2. Binding Protein

circRNAs can also contain specific binding sites for RBPs, a class of proteins involved in regulating gene transcription and translation [44]. The interaction between circRNAs and RBPs can influence various cellular processes, including circRNA stability, protein biogenesis, and the transcriptional regulation of their parental genes [45]. A well-characterized example of this mechanism is circANRIL encoded at the 9p21 locus, which regulates ribosome biogenesis and contributes to atheroprotection [46]. CircANRIL physically binds to the PES1 protein, an essential factor for 60S pre-ribosomal RNA maturation. This binding is mediated by a conserved secondary structure within circANRIL—specifically, an RNA fork or stem-loop—that structurally mimics pre-ribosomal RNA and acts as a competitive decoy [47].

### 3.3. Protein Scaffold

circRNAs can regulate gene expression and protein functions by facilitating protein–protein interactions, acting as molecular scaffolds [48]. An example is circFOXO3 that can promote apoptosis in cancer cells, interacting with the protein p53 and the protein MDM2 [49]. In the cardiovascular system, circ-Amolt1 acts as a scaffold molecule that physically interacts with both AKT and PDK1, promoting AKT phosphorylation and subsequent nuclear translocation. This translocation is crucial for cardiomyocyte survival and proliferation, offering a protective effect against doxorubicin-induced cardiomyopathy [50].

### 3.4. Protein Translation

Although circRNAs were long considered non-coding due to the absence of a 5′ cap in their transcripts [51], it is now well established that circRNA translation occurs via cap-independent mechanisms. These require the presence of an open reading frame (ORF) that ribosomes recognize to synthesize peptides [52]. Moreover, the mechanism involves internal ribosome entry sites (IRESs)—structured RNA elements—present within circRNAs that recruit the 40S ribosomal subunit independently from the 5′ cap, facilitating the translation initiation [53]. This process also depends on canonical translation initiation factors (eIFs) and other trans-acting factors, like those involved in linear mRNA translation [54].

Indeed, various IRES-like elements supporting cap-independent translation have been identified in circRNAs, including sequences complementary to 18S rRNA, specific RNA secondary structures, hexamer motifs, and poly(U) tracts [55]. Additionally, repeat-associated non-AUG (RAN) translation occurs at near-cognate codons, adjacent to expanded RNA repeats, enabling non-canonical protein synthesis from circRNAs [56]. Another cap-independent translational mechanism involves N^6^-methyladenosine (m^6^A). This epi-transcriptomic modification regulates various aspects of mammalian RNA metabolism, including translation [50]. A notable example is circZNF609, a bifunctional circRNAs that encodes a protein through a splicing-dependent IRES and whose translation is modulated by m^6^A modifications via interactions with m^6^A reader proteins. Functionally, circZNF609 plays crucial roles in myogenesis, tumorigenesis, and tissue repair, with significant implications in oncology and cardiovascular biology [57,58,59].

### 3.5. Regulation of Gene Transcription

ciRNAs and EIciRNAs are predominantly localized in the nucleus, where they regulate transcription of their parental genes (cis effect) [60,61]. Both classes directly interact with RNA polymerase II or U1 small nuclear ribonucleoprotein (snRNP) complexes to modulate transcription [62]. Ci-ankrd52, originating from intron 2 of the *ANKRD52* gene, is one known example. Ci-ankrd52 accumulates near its transcription site and binds phosphorylated Pol II (involved in the transcript elongation), enhancing transcriptional efficiency, facilitating R-loop resolution, and promoting RNA polymerase progression. Studies of ci-ankrd52 knockdown demonstrated that *ANKRD52* expression was reduced without affecting neighboring genes, indicating a cis-acting regulatory mechanism [15]. Similarly, circEIF3J, an EIciRNA containing both exonic and intronic sequences presenting the U1 snRNA binding sites, forms a complex with U1 snRNP that associates with Pol II near the *EIF3J* promoter, enhancing transcription initiation. Disruption of this interaction, blocking the U1 binding site via morpholino oligonucleotide antisense, for instance, decreases Pol II recruitment and gene expression, supporting its role as a transcriptional scaffold [31,63].

## 4. Methodologies for CircRNA Detection and Quantification

CircRNAs are indeed expressed in a tissue- and cell-specific manner [64,65]. Although their expression is not directly proportional to their linear counterparts, due to the likely competition mechanism in biogenesis, the level of expression of circRNAs is generally lower. Interestingly, the accumulation of these RNAs depends on the cellular proliferation rate and their biological functions [66]. Rapidly dividing cells show lower circRNAs levels, while in stable, low-proliferating cells—such as neurons and cardiomyocytes—circRNAs accumulate in the cells and often exceed linear transcript isoforms [67,68].

CircRNA identification is based on the detection of the BSJ. This unique sequence is specific to the circular molecule that originates during the circularization of RNA and permits the unequivocal recognition of circRNAs from their linear transcript [69].

To facilitate circRNA detection and quantification, samples can be treated with RNase R, an exonuclease that preferentially degrades linear RNAs, enriching circular species [70]. However, RNase R treatment is not always absolute: some circRNAs are susceptible to degradation, while certain linear transcripts with stable secondary structures may resist enzymatic digestion [71]. Notably, the RNase R treatment also serves as a validation step for circRNA circularity status.

### 4.1. Detection of CircRNAs

CircRNAs can be identified using various high-throughput techniques, which are summarized in Table 1 and briefly described below:RNA sequencing (RNA-seq) represents an untargeted strategy to identify the whole RNA transcript, including the de novo identification of circRNAs [64]. However, due to the generally low expression levels of circRNAs, RNA-seq requires very high sequencing depth and coverage. Read lengths greater than 100 bp facilitate accurate alignment and detection of BSJs and are preferred for this type of investigation [69]. Due to the complexity and lack of standardization in bioinformatic pipelines for circRNA detection, well-structured and robust computational workflows are essential to ensure accurate and reproducible data analysis.Microarray assays represent a targeted methodology for profiling circRNA expression by using probes designed to specifically hybridize the known BSJs that are unique to each circRNA. This high-throughput approach provides greater sensitivity and more accurate detection of known circRNAs compared to sequencing approaches, because it specifically amplifies the signal for known targets, making it particularly indicated for low-abundance species. However, microarrays are limited only to previously discovered circRNAs and do not simultaneously quantify the linear transcripts, limiting the direct comparative analyses of circular vs. linear isoform quantity [72].

However, both RNA-seq and microarray-based methods rely on reverse transcription, which might introduce errors, including false positive results due to template switching and ligation artifacts [73,74]. Moreover, concatemer formation and trans-splicing events [75] can alter circRNA detection [76]. These technical limitations, although rare, must be carefully considered during data interpretation.
3.Long-read sequencing technologies, another important aspect to consider during circRNA investigation, is the length of sequencing reads, especially when multi-exonic circRNAs are present. Long-read sequencing technologies, such as Oxford Nanopore Technology (ONT), provide a suitable approach to overcome these limitations by enabling full-length circRNA sequencing and improving the resolution of complex splicing isoforms, including multi-exonic circRNAs [77,78]. In addition, this method facilitates the identification of novel circRNAs derived from gene fusions and read-through transcripts, providing a more comprehensive and accurate circRNA landscape [79].4.Single-cell RNA sequencing and spatial transcriptomics: Recent advances in sequencing technologies, such as single-cell RNA sequencing and spatial transcriptomics, enable the profiling of circRNA expression at single-cell resolution and subcellular resolution within tissue architecture, respectively. These approaches overcome limitations inherent to bulk RNA-seq which analyzes tissue samples, often composed of heterogeneous cell populations and cell-type-specific circRNA expression patterns. Both techniques allow precise identification of circRNA distribution across subcellular compartments (including the nucleus, cytoplasm, and mitochondria), providing detailed spatial and temporal information on circRNA expression [68,80,81]. However, their high cost and technical complexity currently limit their widespread use.
ijms-26-09725-t001_Table 1Table 1Methodological approaches for circRNA detection and validation, benefits and drawbacks.MethodologyCostProsConsRefsGenome-Wide ApproachesMicroarray assay+++Wide characterization High identification efficiency Standardized bioinformatic analysis No de novo identification No differential expression Prone to false positive (reverse transcription + amplification) [69,73,74,75,76]RNA-sequencing+++Genome-wide characterization De novo identification Differential expression analysis High sequencing depth required Complex Prone to false positives (reverse transcription + amplification) [72,73,74,75,76]Long-read sequencing+++Full-length sequencing Ideal for multiexonic and splicing isoform detection Fusion- and read-through-derived circRNA detection Complex Specific technology [77,78,79]Single-cell sequencingSpatial transcriptomics++++Overcome RNA-seq biases Single cell/subcellular compartment resolution Spatial and temporal data Complex High expertise Specific technology [68,80,81]Locus-specific approaches: PCR-basedRT-PCR+Splicing isoform detection BSJ sequence verification Qualitative detection Intronic circRNAs undetectable Prone to false positives (retrotranscription + amplification) [73,74,75,76,82]RT-qPCR++Quantitative assessment Relative quantification (circ to linear ratio) Splicing isoform detection Intronic circRNAs undetectable Prone to false positives (retrotranscription + amplification) [83,84]ddPCR++Absolute quantification Less susceptible to artifacts Ideal for low-concentration samples (e.g., plasma/serum) Intronic circRNAs undetectable Specific technology [85,86]Locus-specific approaches: Hybridization-basedNorthern blot+PCR-free method Enzyme-free detection High level of circRNA expression required Large quantity of starting material Time-consuming [87,88]NanoString nCounter++PCR-free method Single molecule counting Ideal for low-quality samples (e.g., FFPE) Specific technology Large quantity of starting material Time-consuming [89,90,91]In situ hybridization++PCR-free method Subcellular circRNA localization High level of circRNA expression required Large quantity of starting material Time-consuming [92,93]BSJ: back-splice junction; ddPCR: droplet digital polymerase chain reaction; FFPE: formalin-fixed paraffin-embedded; RT-PCR: reverse transcription polymerase chain reaction; RT-qPCR: real-time quantitative polymerase chain reaction; + to ++++: increasing levels of cost.

### 4.2. Challenges in Data Analysis for CircRNA Detection

Although NGS technologies are powerful tools for discovering and detecting circRNAs, the analysis of the large volumes of data they generate presents significant challenges. Effectively addressing these tasks requires careful methodological considerations and robust expertise in bioinformatics and data analysis. The main computational strategy for circRNA detection is the identification of the BSJ, as previously described. However, several challenges complicate circRNA identification, including the following:Alternative splicing and exon selection can generate multiple isoforms from a single gene locus. Consequently, linear isoforms and circRNAs can share sequence regions, making it difficult to computationally distinguish between them [77].CircRNAs can also have multiple isoform variants produced by differential exon selection or alternative BSJs, making it relevant to identify the exon composition of each circular isoform [94,95]. A simple counting of BSJ-spanning reads can misestimate circRNA abundances due to isoform overlap and shared sequences.If a gene, or parts of it, containing splice sites is duplicated in other genomic locations, sequencing reads may align ambiguously. This can create apparent “jumps” between homologous regions that mimic BSJs [74,96]. Additionally, transcripts from inverted or repeated regions can also produce signals resembling BSJs because of reverse-oriented splicing [96].Technical artifacts may further complicate analysis. For example, template-switching during reverse transcription can generate chimeric cDNA molecules that falsely appear as back-splicing events, especially when homologous regions are in proximity [73,97]. It has been estimated that 35–55% of computationally detected isoforms for a gene result from this artifact [98].Ultimately, sequencing errors, particularly at exon boundaries and in genes with high sequence similarity, can also introduce false splice signals (e.g., erroneous GU/AG) and need to be carefully considered [99].

Importantly, since circRNAs are generally expressed at lower levels than other RNA species, simply filtering by read depth is insufficient to remove these artifacts. To address these complexities, most bioinformatic tools use paired-end sequencing data, where one read spans the BSJ and its paired read maps within the circular transcript at a defined distance, improving specificity and accuracy in circRNA identification [82].

### 4.3. Computational Identification of CircRNAs

Tools for circRNA detection fundamentally rely on the identification of BSJs through alignment strategies capable of recognizing non-colinear mapping fragments indicative of back-splicing events. circRNA levels are subsequently quantified by counting the reads that align at their BSJs. Given that many reads originate from spliced transcripts, all circRNA identification tools depend on splice-aware aligners, such as STAR [100], HISAT2 [101], Bowtie2 [102], TopHat2 [103], segemehl [104], and MapSplice [105], to accurately map RNA sequencing reads to the reference genome.

Algorithms for circRNA detection can be divided into two main categories: split-alignment-based and pseudo-reference-based approaches. Both seek to identify BSJs but differ significantly in how sequencing reads are mapped and analyzed.

Split-alignment-based tools detect BSJs by identifying chimeric or discordant reads that are not mapped continuously to the reference genome. These reads are split into segments and mapped to two separate genomic regions in reverse orientation, consistent with back-splicing events. Popular tools in this category include find_circ [67] circRNA_finder [106], CIRI2 [107], *DCC* [108], and CIRCexplorer2 [94] (Table 2). In contrast, pseudo-reference-based methods construct artificial reference sequences by compiling or collecting a custom library of manually curated circular junctions, to which sequencing reads are then aligned. Tools based on this approach comprise KNIFE [109], Acfs [110], and PTESFinder [111].

The principal distinction between these two methodological strategies lies in their de novo discovery capabilities and identification sensitivity. Split-alignment-based tools are often preferred for their high sensitivity and ability to discover novel circRNAs but are more susceptible to false positives caused by mapping errors, repetitive regions, or technical artifacts. Therefore, rigorous data filtering based on read quality, splice site motifs, and alignment confidence is crucial for improving the detection of true circular junctions by these methods. Conversely, pseudo-reference-based approaches generally offer higher accuracy, but are limited to the identification of annotated circRNAs, restricting their discovery potential. Balancing this trade-off between discovery power and specificity, alongside considering the research context and scientific goals, is fundamental for choosing the appropriate method in circRNA data analysis.

#### 4.3.1. Combination of Methods Limit Tool Variability

The variability in approaches used by circRNA detection tools also results in considerable differences in predictions, even when applied to the same dataset. Comparative studies have shown modest overlap between circRNAs predicted by different tools, with many circRNAs identified by only one algorithm. For instance, Hansen et al. (2015) [128] demonstrated that out of five tools tested (circRNA_finder, MapSplice, CIRCexplorer [9], CIRI [32], and find_circ), only 16% of BSJs were predicted by at least two methods. In 2018, Hansen et al. [129] replicated the analysis using 11 tools (Acfs, CIRCexplorer, CIRCexplorer2, CIRI, CIRI2, DCC, find_circ, KNIFE, MapSplice and UROBORUS [112]), confirming similar results to the 2016 study, and showing that the combination of more tools significantly reduces false positive rates and mitigates individual tool weaknesses.

Considering these findings, integrated strategies have gained popularity, and integrative pipelines have been developed. Examples of pipelines implementing a multi-tool approach for circRNA detection are CirComPara2 [113], circRNAwrap [114], and nf-core/circrna [115]. CirComPara2, for instance, integrates up to nine computational pipelines built by combining different circRNA detection tools and aligners. This approach allows users to customize their data analysis by choosing the most appropriate combination of tools. Similarly, circRNAwrap incorporates eight detection tools and three reconstruction tools, providing both detection and structural characterization within a single platform, which is particularly useful for downstream functional investigations. Meanwhile, nf-core/circrna supports up to seven BSJ detection methods and extends functionality to differential expression analysis and circRNA-miRNA interaction prediction. The main advantage of these integrated pipelines lies in their ability to filter out method-specific noise by requiring consensus across multiple algorithms. For example, CirComPara2’s approach of retaining only BSJs supported by at least two tools has been shown to substantially reduce false discovery rates [113]. At the same time, the inclusion of tools with varying stringency ensures that the pipelines capture both common and rare circRNAs.

#### 4.3.2. Machine Learning for CircRNA Prediction

Despite integrative approaches improving accuracy at the cost of sensitivity in circRNA detection, distinguishing real circRNAs from other species of non-coding RNAs remains challenging. Several machine learning (ML) models have been developed to predict circRNAs from other classes of RNA based on sequence and structural features, proving to be efficient and valuable tools for this purpose. The biogenic characteristic signatures generally used by ML models are specific biological signatures that favor circRNA formation. Among the most predictive characteristics are ALU elements and tandem repeat motifs in flanking intronic regions, which facilitate the spatial proximity of distal splice sites through the formation of hairpin secondary structures [6]. These structural elements, combined with evolutionary conservation scores and SNP density within conserved miRNA binding sites [130], constitute the biological features upon which circRNA predictive models are generally built. Early approaches, such as PredcircRNA [122] ranked 188 features by their importance based on Random Forest (RF), and identified conservation features, GT/AG motifs, and component composition scores as the most important predictors for circRNAs. Building on this, circDeep [123] introduced a deep learning (DL) model that integrates local sequence context with the presence of reverse complementary matches (RCMs) and conservation patterns. This integrative strategy improved circRNA prediction accuracy by 12% compared to PredcircRNA. Other tools like DeepCirCode [124] and JEDI [125] moved beyond general classification to predict the likelihood of back-splicing events through learning interdependencies between splice sites. DeepCirCode trained a convolutional neural network (CNN) using 50 nt-long sequences flanking BSJs to identify interdependencies between splice junctions with a high probability of producing circRNAs. Unlike traditional classifiers, JEDI can predict novel circRNAs without prior BSJ annotations. It does not restrict its analysis to just two splice sites, as DeepCirCode does; rather, it models the sequence context and flanking regions of all potential splice sites within a transcript. JEDI is particularly suited for de novo circRNA prediction, making it useful for discovering new circRNAs in novel transcripts across different tissues and conditions. More recently, circCNNs [126] introduced CNN architecture specifically designed to understand exonic circRNA biogenesis. Using a dataset of curated pairs of back-splicing and linear-splicing exons, circCNNs integrates junction sequence data and RCM-extracted features to identify motifs recognized by known back-splicing regulators such as MBNL1, QKI, and ERSP2, achieving a specificity greater than 90%. Notably, applying RCM metrics significantly reduced false positives and highlighted the importance of structural information in correctly identifying BSJ sites with an overall prediction accuracy of 88%. Interestingly, a recent tool called CIRI-deep employs a deep neural network trained with over 25 million BSJ events derived from bulk RNA-seq data [127], integrating a broad range of cis- and trans- features to identify circRNAs and detect differentially spliced circRNAs between tissues or experimental conditions, even from spatial transcriptomic and single-cell data. Nevertheless, the model is trained on normal samples, which limit its applicability in disease conditions.

#### 4.3.3. Reconstruction of Full-Length CircRNAs

Identifying BSJs is only the first step toward understanding circRNA function. Most circRNA detection tools stop at BSJ localization without providing information about the full exon composition of the circular transcript. However, full-length reconstruction is critical for annotating miRNA-binding sites, RBP motifs, and coding potential.

Among these, CIRI-full [116] extends the output of CIRI2 by integrating exon usage, estimated by the CIRI-AS tool [26], and paired-end read information to reconstruct circular transcripts from BSJ coordinates.

The CIRCexplorer suite also attempts full-length circular reconstruction. While CIRCexplorer2 already supports de novo assembly and junction refinement for full-length reconstruction, the more recent CIRCexplorer3 [117] represents a significant advancement. It integrates splice site coverage information from both poly(A)+ and RNase R-treated RNA-seq libraries, providing a more accurate and comprehensive reconstruction of circRNA isoforms. CIRCexplorer3 can identify exon skipping, alternative 5′ and 3′ splice sites, and retained introns within circular transcripts, making it particularly powerful for studying the isoform diversity and regulatory complexity of circRNAs.

Additional tools such as circAST [118], psirc [120], and CYCLeR [95] have introduced splice-graph-based or statistical inference approaches to model exon connectivity in circular transcripts.

Complementing these short-read-based methods, third-generation sequencing platforms have enabled the direct sequencing of full-length circRNAs. Multiple tools, including circFL-seq [119], isoCirc [121], and CIRI-long [76] are tailored to capture full-length circular isoforms, including those containing microexons or regulatory motifs. Although long-read platforms still present higher error rates and costs compared to second-generation sequencing, they provide a valuable resource for validating and refining reconstructions derived from short-read data.

#### 4.3.4. CircRNA Databases

A comprehensive understanding of circRNA biology and its roles in disease mechanisms relies on access to stable and well-curated databases as well as the construction of regulatory interaction networks. For example, circBase [131] is one of the earliest and most widely used resources, aggregating circRNA data from diverse sources and providing genomic coordinates, sequences, and expression evidence. Similarly, CircAtlas [132] and CIRCpedia [133] present extensive annotations, with circAtlas focusing on cross-species conservation and tissue-specific expression profiles, while CIRCpedia provides functional annotations and information on splice sites. CircNet [134] gathers data from over 400 samples across 26 diverse tissues and offers valuable insights into circRNA expression and the competing endogenous RNA regulatory networks involving circRNAs, miRNAs, and genes. However, one of the persisting challenges in the field is the lack of consistency in circRNA naming conventions across the databases, which hampers the ability to cross-reference circRNAs between resources. To address this, CircAtlas 3.0 [135] introduced enhanced features, including mapping tools to convert identifiers from major databases such as circBase, circBank [136], and CIRCpedia, promoting a common nomenclature system. For functional annotation and interaction data, databases like starBase v2.0 [137] and CircInteractome [138] are essential. StarBase leverages CLIP-seq data to infer RNA-RNA and RNA-protein interactions across up to 23 species. CircInteractome, with a human-centric focus, provides tools for predicting miRNA and RBP binding sites, siRNA design, and divergent primer generation. Databases specifically cataloging disease-associated circRNAs are also crucial. Circ2Traits [139] was the first resource to integrate manually curated associations between circRNAs, miRNAs, and disease phenotypes. Other databases have also been developed, such as Circ2Disease [140], CircRNADisease [141], LncRNADisease 3.0 [142], and CircR2Disease [143], which offer experimentally validated circRNA–disease associations. Another relevant resource is circRNADb [144], which contains annotations of exonic circRNAs with coding potential.

In summary, while these databases collectively advance circRNA research and have complementary functionalities, the field continues to face challenges related to integration and standardization, particularly in naming conventions and reference annotations. Addressing these issues is essential to ensure greater consistency and reliability in circRNA studies.

### 4.4. Validation and Quantification: Locus-Specific Approach

Given the diversity in circRNA biogenesis, sequence composition, and expression patterns, combined with the inherent limitations of each detection method, a gold standard approach for circRNA detection currently does not exist. Therefore, experimental validation of initial circRNA findings is essential, often using established protocols adapted from studies of other non-coding RNAs, such as miRNAs. 

Validation of circRNAs specifically targets their unique BSJ, not present in linear isoforms. Accurate knowledge of the BSJ genomic coordinates is necessary to properly guide downstream experimental design.

#### 4.4.1. PCR-Based Methods

PCR is a widely used validation method for circRNAs due to its quickness and low input requirements. The key strategy involves designing divergent primers that flank the BSJ, allowing selective amplification of circular molecules. This method relies on the fact that amplification occurs only when exons are joined in inverse order, specifically avoiding amplification of the linear form. Three PCR based methods are commonly used for this purpose:Qualitative detection of circRNAs and their splicing isoforms. In this approach reverse transcription PCR (RT-PCR) is followed by gel electrophoresis and Sanger sequencing to confirm probe specificity to the BSJ sequence [82].Quantitative assessment through real-time quantitative PCR (RT-qPCR) allows relative quantification of circRNAs and optionally, linear transcripts by combining a BSJ –specific assay with additional primers targeting linear exon junctions. Melting curve analysis and gel electrophoresis can also identify alternative splicing isoforms [83,84,145].PCR-based methods involving reverse transcription can induce template-switching artifacts, generating false positives. Additionally, they cannot assess intronic circRNAs since primers target exon junctions specifically. Furthermore, rolling circle amplification (concatemer formation) and trans-splicing events, might complicate quantification [76,84,89].Digital droplet PCR (ddPCR) offers absolute quantification with improved specificity and sensitivity. Even though it is based on PCR technology, it is less susceptible to artifacts introduced during reverse transcription and is particularly indicated for samples with low circRNAs quantity, such as plasma or serum [85,86].

#### 4.4.2. Hybridization-Based Methods

Similarly, these technologies rely on the identification of the BSJ sequence by using probes instead of primers. The most used methods are listed below:Northern blotting was used in early circRNA studies. It enables visualization of circRNAs on agarose gels using BSJ-specific probes after RNase treatments that confirm circularity. It is indicated for circRNAs with relatively higher expression levels [87,88].NanoString nCounter, which requires specific instrumentation, utilizes fluorescent probes targeting BSJ sequences, and through streptavidin-biotin interactions enables detection down to the single-molecule level. This technology is convenient for low-quality samples, like FFPE (formalin-fixed, paraffin-embedded) tissues [89,90,91].In situ hybridization (ISH) using fluorescent or chromogenic probes enables precise localization of circRNAs within subcellular compartments through microscopic visualization [92,93].

In all these approaches, RNase and protease pre-treatments help to reduce background signals from linear RNAs and associated proteins, improving visualization. Although hybridization-based methods provide valuable spatial and molecular information, they require large amounts of starting material and are time-consuming. For these reasons, visualization of circRNAs using hybridization methods is less common. Indeed, spatial information on circRNA expressions is extremely useful for elucidating their cellular functions [146]. Therefore, validation remains fundamental, offering diverse and complementary insights. An ideal validation strategy for circRNAs would integrate both PCR-based and hybridization-based methods, ensuring more robust and comprehensive confirmation of circRNA data.

### 4.5. Functional Characterization

Functional studies are essential to elucidate the cellular roles and interaction partners of circRNA independently of their linear RNA isoforms. These experiments aim to determine how specific circRNA expression patterns influence cellular phenotypes, potentially through modulation of gene functions that remain to be fully understood [69]. To this purpose, loss-of-function and gain-of-function methods are indicated for circRNAs’ functional characterization:Loss-of-function studies primarily use RNA interference techniques such as siRNAs or antisense oligonucleotide (ASOs) specifically designed to target the back-splice junction (BSJ) sequence unique to circRNAs [69]. Targeting the BSJ allows for selective knockdown of circRNAs without affecting linear isoforms. Given the possibility of off-target effects in RNA interference experiments, it is recommended to design multiple siRNAs targeting different sequences of the BSJ and to include appropriate controls such as scrambled siRNAs. To accurately verify the knockdown efficiency and ensure phenotype specificity, the circRNA to linear RNA ratio should be consistently monitored through the experiment [147].Additionally, CRISPR-Cas-based genome editing tools are increasingly employed for circRNA depletion with Cas9 targeting DNA genomic loci [148], and Cas13 acting directly on RNA through sequence specific cleavage [149]. The CRISPR-Cas9 system allows circRNA depletion by deleting the entire gene locus [150], an approach applicable when the corresponding linear RNA is minimally or not co-expressed, or by removing specific DNA elements that promote circRNA biogenesis [2,151]. These elements include splice sites and repetitive sequences, crucial for back-splicing (see Section 2). Accurate design is essential to minimize off-target effects and to preserve expression of the linear mRNA counterpart. Efficient depletion is confirmed by PCR based methods, verifying an approximately 50% reduction in circRNA expression, consistent with allele-specific deletion. An alternative method to deplete circRNA expression, which acts directly on RNA instead of DNA and is less drastic than genome editing, is RNA-directed RNA-cleavage, obtained by using the CRISPR-Cas13 system [149,152]. The guide RNA in this system is designed to overlap the BSJ of the circRNAs, driving specific RNA recognition and cleavage of the circRNA molecule. However, optimization of the guide RNA design and targeting is recommended to minimize potential off-target effects such as cleavage of the corresponding linear RNA. Nevertheless, RNA-directed RNA-cleavage has been proved to be more specific compared to RNAi-mediated depletion [153,154].CircRNA depletion typically leads to observable phenotypic changes that reflect its functional role, including alterations in gene expression profiles, cellular behaviors, or developmental processes. For example, in vivo knockdown studies have shown that circRNA depletion causes transcriptome changes, impacting specific biological pathways or phenotypes, such as altered neuronal activity, sensory functions, or tumor-related traits. Thus, these methods allow the assessment of phenotypic outcomes directly attributable to circRNA loss, providing insights into their biological and translational significance [147].Gain-of-function experiments (rescue experiments), used to determine circRNA’s role in a biological process, are performed following circRNA inhibition to confirm that observed cellular phenotypes are directly linked to circRNA depletion. These approaches involve the exogenous expression of circRNAs resistant to RNAi or CRISPR-Cas13 cleavage, often delivered via plasmids or viral vectors. [28,155]. With these overexpression methods, circRNAs can also be investigated to study their effect on cells. circRNAs can also be synthetized in vitro (IVT) using different methods [156,157] and subsequently transfected into cells. The advantages of the IVT circRNAs are related to the fact that they offer precise control over length, circ-to-linear ratio and overall quantity. However, co-transcriptional modification such as m^6^A methylation and interaction with RNA-binding proteins (RBPs) may be absent in IVT circRNAs, potentially affecting their biological role [158]. To ensure specificity, IVT circRNAs should be treated with RNase R and highly purified before transfection. In contrast, in vivo expression systems use cellular splicing machinery to generate the circRNAs. Functional sequence elements, including splice sites and intronic inverted repeated sequences (natural or artificial), are incorporated into expression plasmid [11,159,160]. These plasmids undergo back-splicing to circularize the transcript. This method is usually less efficient than IVT, often producing linear transcripts, read-through, or concatemeric RNA species alongside circRNAs. Therefore, it is important to verify the circularity using RNase R treatments, monitor circ-to-linear ratio and confirm the correct BSJ formation throughout overexpression experiments [161,162,163,164]. Delivery of circRNA-overexpressing plasmids is typically achieved using viral vectors, such as adenovirus-associated virus (AAV) [165].

#### 4.5.1. CircRNA-miRNA Interaction

CircRNAs play diverse functional roles in cells, influencing various biological processes primarily through interactions with other biomolecules [28,155]. Among these, miRNA sponging is the most extensively studied function of circRNAs, especially in cardiovascular research. However, as the earliest investigated role, it may have introduced a “primary role bias” in subsequent studies, suggesting that all circRNAs predominantly act as miRNA sponges. Importantly, experimental validation of circRNA-mediated miRNA sponging is complex and requires a multi-step approach [166]. Initially, bioinformatic analyses are performed to identify putative miRNA binding sites, known as MREs, within circRNA sequences. Prediction of MREs is based on algorithms such as miRanda [167], TargetScan [168], or RNAhybrid [169], which consider sequence complementarity, thermodynamic stability, and evolutionary conservation. These results are often integrated with experimentally validated interactions from circRNA databases to build competing endogenous RNA (ceRNA) networks. However, many circRNAs harbor no more MREs than expected by chance, and the simple presence of MREs does not prove functional interaction [170]. The stochiometric ratio of MREs in circRNAs and target mRNA must be evaluated when studying miRNA’s sponging function, considering also that the detected interaction may depend on other features like secondary structure, subcellular localization, and others [39].

To experimentally validate circRNA–miRNA binding and crosslinking, Argonaute (Ago) immunoprecipitation assays are conducted to isolate the RNA-induced silencing complex (RISC) along with its associated RNAs [171]. Subsequent quantification of circRNAs and miRNAs in immunoprecipitated samples is performed using reverse transcription quantitative PCR (RT-qPCR). After confirming physical interaction, functional validation involves evaluating changes in endogenous miRNA target gene expression upon circRNA modulation, typically through circRNA overexpression or knockdown experiments. This integrated approach provides robust evidence for the functional relevance of circRNA–miRNA interactions [39,172].

#### 4.5.2. CircRNA–Protein Interaction

CircRNAs can function as “sponges” or scaffolds for proteins, facilitating protein interactions. However, demonstrating that these interactions are specific to circRNAs rather than their linear isoforms remains challenging. Differences in structure, subcellular localization, and stability between circular and linear RNAs suggest that distinct protein complexes may be generated, despite both sharing most of their coding sequences [173]. CircRNA–protein interactions can be investigated using either a circRNA-centric approach or a protein-centric approach. In both cases, complexes between the target circRNA and interacting proteins are analyzed following immunoprecipitation (IP).

In the circRNA-centric approach, IP is performed using specific antisense oligonucleotides complementary to the BSJ sequence [45,174]. The proteins associated with the precipitated complex are then isolated and analyzed by Western blotting or mass spectrometry, the latter being particularly useful for unbiased protein identification [175].In the protein-centric approach, the target protein within the ribonucleoprotein (RNP) complex is immunoprecipitated, followed by detection and quantification of associated circRNAs through RNA sequencing or direct quantification methods as described previously [176].

It is necessary to validate pull-down efficiency by immunoprecipitating endogenous RNP complexes from the same cell lysates. Furthermore, specificity should be confirmed by verifying the absence of the targeted RNP complex in cells lacking expression of circRNAs. Accurate estimation of circRNA abundance within complexes requires monitoring IP efficiency relative to input sample [176].

For reliable interpretation of interactions involving both miRNAs and proteins, careful monitoring of the circRNA-to-linear RNA ratio is critical to correctly assess the relative abundance of circRNAs throughout the experiments [69]. This ensures biologically meaningful conclusions based on the endogenous expression levels. While several circRNAs have been proposed to function as molecular sponges based on predicted binding sites or in vitro interactions, rigorous validation remains necessary due to their typically low endogenous abundance. Further evidence of circRNA–protein or circRNA-miRNA interaction can also be obtained through complementary visualization methods such as in situ hybridization (ISH) to confirm co-localization of circRNAs with their targets [80].

#### 4.5.3. CircRNA Translation

Initially considered non-coding, circRNAs have now been shown to possess protein-coding potential, despite lacking canonical features necessary for translation, such as a 5′ cap and polyadenylated tail [57]. Studying circRNA translation first requires identifying a putative open reading frame (ORF) within the circRNA’s sequence. Due to the absence of the 5′ cap structure, efficient translation initiation depends on the presence of an internal ribosome entry site (IRES) or equivalent ribosome-binding elements upstream of the ORF [52,53,54]. The functionality of these elements must be experimentally validated, typically using reporter assays [57,177,178,179,180]. Although an association of ribosomes with circRNAs can be demonstrated, such interaction does not naturally confirm its effective translation. One approach to study circRNA translation involves exogenous expression via plasmids or transfection with IVT molecules followed by the detection of peptides derived from the circRNAs [69]. However, exogenous expression systems, especially plasmids, often produce both circular and concatenated linear RNA isoforms, complicating the attribution of peptides exclusively to circRNA-derived translation [163,181]. Definitive identification of circRNA-encoded peptides relies on the detection of sequences spanning the BSJ and extending to the translation termination codons. Moreover, peptide abundance in exogenous circRNA constructs may not reflect endogenous levels [163,181].

Detection of circRNA-derived proteins is commonly performed using Western blotting with antibodies targeting predicted peptide epitopes or via mass spectrometry. The incorporation of molecular tags through gene editing can facilitate peptide detection and enable the study of interacting protein partners [57,177,178,179,180,181]. The functional relevance of circRNA-encoded peptides is assessed through modulation of circRNA expression and evaluation of consequent cellular phenotypic alterations [69]. However, the discovery of circRNA translation products remains challenging due to the generally low endogenous circRNA expression resulting in limited peptide yield [182].

Notably, circRNA translation may be regulated by cellular context, including external stimuli and stress conditions, highlighting the need for careful experimental design to elucidate their physiological relevance [57,178].

## 5. Role of CircRNAs in Cardiovascular Diseases

CVDs remain the leading cause of mortality worldwide, and with population aging, their incidence is expected to increase further (World Health Organization-CVDs 2025). CircRNAs, along with other non-coding RNAs, have been increasingly implicated in the pathogenesis of various complex diseases including cancers [183], neurodegenerative disorders [184], and importantly, CVDs [185]. These CircRNAs participate in intricate regulatory networks across diverse cardiac cell types such as cardiomyocytes, cardiac fibroblasts, endothelial cells, and vascular smooth muscle cells. The dysregulation of circRNAs in these cellular populations contributes to the pathological vascular and myocardial remodeling observed in various cardiovascular conditions [186].

Among the most studied CVDs in circRNA research are cardiac hypertrophy, fibrosis, heart failure, myocardial infarction, coronary artery disease, and atherosclerosis, as can be seen in Table 3, as circRNAs are reportedly involved in these pathologies.

Metabolic syndromes, including diabetes as well as cerebrovascular events such as stroke, aneurysms, and pulmonary complications, have also been investigated for circRNA involvement [187]. Functionally, the predominant mechanism attributed to circRNAs in CVDs is miRNA sponging, wherein circRNAs modulate RNA interference pathways by sequestering specific miRNAs. Emerging evidence also suggests the existence of circRNA-derived micropeptides with potential regulatory roles, though experimental validation, such as for circZNF609, remains limited [57,58,188].

CircRNAs participate in a wide array of cellular processes critical to cardiovascular homeostasis and disease progression. These include the regulation of programmed cell death modalities such as apoptosis [42,47,171,189,190,191,192,193,194,195,196,197,198,199,200,201,202,203,204,205], autophagy [42,191,192,193,206], modulation of mitochondrial function and dynamics [207,208,209], and involvement in cell proliferation signaling pathways [47,175,189,210,211,212,213,214,215,216,217,218,219,220]. Moreover, circRNAs participate in cellular stress responses that promote survival [50,221,222,223], pathological differentiation [224,225,226], and cellular senescence [227,228,229].

Some of the best functionally characterized circRNAs, as reported in cardiovascular research studies, are listed in Table 3. Though much data is available in the literature, consistent validation is still lacking.
ijms-26-09725-t003_Table 3Table 3Examples of well characterized circRNAs in cardiovascular pathophysiology. This table summarizes experimental evidence regarding circRNAs identified in cell culture models, animal models and human samples. For each circRNA, the relevant sample and pathological context are reported, together with the main molecular mechanism of action, the biological effects observed, and the corresponding references.CircRNA SamplesPathological ContextMechanism of ActionBiological EffectsRefsApoptosis circNCX1/Slc8a1 animal model: TAC mouse; sample: mouse CMs CH/HF miRNA sponge: miR-133a/CTGF-SRF-ADRB1-PRKACB-ADCY6 Induces CH [205]circNCX1/Slc8a1 animal model: MI mouse; sample: Rat CMs MI/I-R injury miRNA sponge: miR-133a-3p/CDIP1 Induces CM apoptosis [171]circNfix animal model: MI mouse/rat; sample: mouse/rat CMs MI/I-R injury miRNA sponge: miR-214/Gsk3β;protein scaffold: Ybx1-Nedd4l Inhibits angiogenesis and CMs proliferation [204]circNfix animal model: TAC mouse; sample: Ang II-treated CMs, human plasma CH/HF miRNA sponge: miR-145-5p/ATF3 Reduces CH [203]HRCR (circ_012559) animal model: ISO-injected mouse, TAC mouse; sample: mouse heart CH/HF miRNA sponge: miR-223/ARC Reduces CH [202]CDR1as animal model: MI mouse; sample: mouse CMs MI/I-Rinjury miRNA sponge: miR-7a/PARP-SP1 Induces CM apoptosis [201]CDR1as sample: Human CMs, AC16 cell line HF miRNA sponge: miR-135(a/b)/HMOX1 Inhibits CM apoptosis [230]CDR1as animal model: DM mouse; sample: D-glucose-treated CMs DM protein sponge: MST1/LATS2-YAP Induces CM apoptosis [200]circCDR1as animal model: AMI pig; sample: porcine heart MI/I-R injury miRNA sponge: miR-7a Inhibits CM apoptosis [199]ACR sample: hypoxia-treated AC16 cells, CHF patient plasma HF unknown: miR-532 promoter methylation Inhibits CM apoptosis [193]MFACR (circ_016597) animal model: MI mouse; sample: human plasma, AC16 cell line, mouse heart MI/I-R injury unknown: miR-125 gene methylation Induces CM apoptosis [197]circMAP3K5 animal model: DM rat; sample: DCM rat hearts; high glucose-induced H9c2 DM miRNA sponge: miR-22-3p/DAPK2 Induces CM apoptosis [196]circZNF609 animal model: I/R injury mouse; sample: mouse CMs, NRCMs MI/I-R injury protein sponge: YTHDF3-YAP Induces CM apoptosis [195]circHIPK3 animal model: TAC mouse; sample: mouse heart, NMCMs, HEK-293T cell line CH miRNA sponge: miR-185-3p/CASR Induces CH [194]Apoptosis-autophagy ACR animal model: I/R injury mouse;sample: mouse CMs, CHF patient plasma MI/I-R injury/HF protein sponge: DNMT3B/Pink1-FAM65B Reduces CM autophagy and apoptosis [198]circHIPK2 sample: H2O2-injured mouse CMs M injury miRNA sponge: miR-485-5p/ATG101 Induces CM autophagy and apoptosis [192]circPAN3 animal model: I/R injury mouse;sample: human CMs, HEK293 cell line M I-R injury miRNA sponge: miR-421/PINK1 Reduces CM autophagy and apoptosis [191]circHIPK3 animal model: MI/I-R injury mouse; sample: mouse heart, NMVCMs MI/I-R injury miRNA sponge: miR-20b-5p/ATG7 Induces CM autophagyand apoptosis [42]Apoptosis—mitochondrial Dynamics and FunctionMFACR (circ_016597) animal model: I/R injury mouse;sample: mouse CMs M I-R injurymiRNA sponge: miR-652-3p/MTP18 Induces mitochondrial fission and CM apoptosis [208]circZNF609 animal model: DOX-induced CT mouse; sample: mouse CFs/CMs, NRCMs DOX-induced CTprotein sponge: FTO Induces CM apoptosis, ROS production, mitochondrial iron overload [209]circSAMD4 animal model: MI mouse; sample: mouse CMs MI/I-R injuryunknown: VCP mitochondrial translocation/VDAC1 Reduces CM apoptosis and mitochondria oxidative damage [207]Apoptosis-proliferationcircANRIL sample: HEK-293 cell line, human vascular tissue, human PBMCs,human primary ECs/SMCs/adventitial FBsASprotein sponge: PES1/nucleolar stress-p53 Induces apoptosis and inhibits proliferation [47]circHIPK3 sample: OGD/R-treated HCMsMI/I-RinjurymiRNA sponge: miR-124-3p/BAX-BCL2 Induces CM apoptosis and inhibits proliferation [189]Apoptosis-inflammationcircFOXO3 animal model: AMI rats; sample: rat heart, H9c2 cell line MI/I-Rinjuryprotein sponge: KAT7/HMGB1 Reduces CM autophagy and inflammation [206]circSAMD4 animal model: AMI mouse; sample: rat CMs, H9c2 CMs MI/I-RinjurymiRNA sponge: miR-138-5p Induces CM apoptosis and inflammation [190]DifferentiationcircSAMD4 animal model: intravascular calcification mouse; sample: mouse aorta, HAECs, CAC patient plasma VCbiomarker Inversely correlates with vascular calcification [224]CDR1as sample: Human PASMCs PHmiRNA sponge: miR-7-5p/CNN3-CAMK2D Induces osteogenic SMC differentiation and calcification [225]circARHGAP10 animal model: aortic valve calcification mouse; samples: mouse aortic valve, human primary VICs, HEK-293T and COS7 cell lines VCmiRNA sponge: miR-355-3p/RUNX2 Induces VIC osteogenic differentiation and calcification [226]Extracellular Crosstalk-proliferationcircHIPK3 animal model: DM mouse; sample: mouse aorta, MAECs, high glucose-induced mouse VSMCs DMmiRNA sponge: miR-106a-5p/FOXO1-VCAM1 Induces VSMC proliferation and inhibits apoptosis [211]circHIPK3 animal model: mouse; sample: hypoxia-pretreated NMCMs, CMECs MI/I-R injurymiRNA sponge: miR-29a/VEGFA Induces CMVEC proliferation, migration, and tube formation [212]Extracellular Crosstalk-survivalcircHIPK3 animal model: mouse; sample: hypoxia-pretreated NMCMs, CMVECs MI/I-R injurymiRNA sponge: miR-29a/IGF-1 Induces CMVEC survival [221]Proliferationcirc5078 sample: HPAECs, BOECs, PAH patient PBMCsPAHProtein sponge:Caprin-1Reduces HPAEC proliferation, stress granule formation, and mitoribosomal protein translation [220]circFOXO3 sample: mouse CF cell linesnon pat.protein scaffold:CDK2-p21Inhibits CM proliferation [174]circRNA_000203animal model: Ang-II induced CH mouse;sample: mouse heart, Ang-II-treated NMVCsCH/HFmiRNA sponge: miR-26b-5p-miR-140-3p/GATA4Induces CH [219]circCHFR sample: ox-LDL treated human VSMCsASmiRNA sponge: miR-370/FOXO1-Cyclin D1Induces VSMC proliferation and migration [210]circNFIB animal model: MI mouse;sample: mouse CFsCFmiRNA sponge:miR-433/AZIN1-JNK1Inhibits CF proliferation [213]circHIPK3 animal model: MI mouse;sample: NMCMsHF post MI/CFmiRNA sponge:miR-17-3p/ADCY6Induces CF proliferation and migration [214]circHIPK3 animal model: Ang-II induced CF mouse; sample: mouse CFs CF/HF miRNA sponge: miR-29b-3p/a-SMA-COL1A1-COL3A1 Induces CF proliferationand migration [215]circHIPK3 animal model: MI mouse; sample: NMCMs, mouse CMs/ECs/CFs, HCAECs MI/I-R injury miRNA sponge: miR-133a/CTGF; protein sponge: NOTCH1 stabilization Induces CM proliferation (NOTCH1), and HCAEC proliferation, migration, and tube formation (miR-133a) [216]circHIPK3 animal model: mouse; samples: hypoxia-treated primary CFs CF miRNA sponge: miR-152-3p/TGF-β2 Induces CF proliferation, migration, and phenotypic switching [217]circCHFR sample: ox-LDL treated human VSMCs, AS patient serum AS miRNA sponge: miR-214-3p/Wnt3-β catenin Induces VSMC proliferation, migration, and inflammation [218]SenescencecircGNAQ animal model: AS mouse; sample: mouse aorta, mouse heart, HUVECs and HCAEC cell lines ASmiRNA sponge: miR-146a-5p/PLK2 Prevents EC senescence [227]ciPVT1 sample: HUVECs and HCAEC cell lines ASmiRNA sponge: miR-24-3p/CDK4-pRb Prevents EC senescence [228]circFOXO3 animal model: CM mouse; sample: mouse CFs/CMs CMprotein sponge: ID-1/E2F1-FAK Promotes CM senescence[229]SurvivalcircFNDC3B animal model: MI mouse; sample: mouse CMs, mouse cardiac ECs MI/I-R injuryprotein sponge: FUS/VEGF-A Reduces CM apoptosis andpromotes neovascularization[222]CNEACR animal model: I/R injury mouse; sample: NMCs, mouse heart MI/I-R injuryprotein sponge: HDAC7/Foxa2-RIPK3 Induces CM survival[223]circAMOTL1 animal model: DOX-induced CM mouse; sample: mouse CMs, Rat YPEN, and mouse CF cell linesCMprotein scaffold:PDK1-AKT1 Induces CM survival[50]Abbreviations, AMI: acute myocardial infarction; Ang: Angiotensin; AS: atherosclerosis; BOECs: blood outgrowth endothelial cells; CAC: coronary artery calcification; CF: cardiac fibrosis; CFs: cardiac fibroblasts; CH: cardiac hypertrophy; CHF: chronic heart failure; CM: cardiomyopathy; CMs: cardiomyocytes; CMVECs: cardiac microvascular endothelial cells; CT: cardiotoxicity; DCM: Diabetic cardiomyopathy; DM: diabetes mellitus; DOX: doxycycline; ECs: endothelial cells; FBs: fibroblasts; HAECs: human aortic endothelial cells; HF: heart failure; HPAECs: human pulmonary artery endothelial cells; I/R: ischemia/reperfusion; ISO: isoproterenol; M: myocardium; MAECs: mouse aortic endothelial cells; MI: myocardial infarction; NMCMs: neonatal mouse cardiomyocytes; NMCs: non-myocyte cells; NMVCMs: neonatal mouse ventricular cardiomyocytes; NRCMs: neonatal rat cardiomyocytes; OGD/R: oxygen-glucose deprivation/reperfusion; PASMCs: pulmonary artery smooth muscle cells; PBMCs: peripheral blood mononuclear cells; PAH: pulmonary arterial hypertension; PH: pulmonary hypertension; ROS: reactive oxygen species; SMCs: smooth muscular cells; TAC: transverse aortic constriction; VC: vascular calcification; VICs: valve interstitial cells; VSMCs: vascular smooth muscle cells.

### 5.1. Myocardial Infarction (MI) and Ischemia/Reperfusion (I/R) Injury

Myocardial infarction is a leading cause of cardiovascular mortality. Ischemia/reperfusion injury causes loss of heart cells such as cardiomyocytes and can eventually lead to heart failure. These processes involve critical cellular functions including programmed cell death, oxidative stress response, and cardiac tissue remodeling, all of which are regulated in part by circRNAs acting as central molecular regulators [231,232,233,234,235].

As summarized in Table 3, many circRNAs function mainly by sponging miRNAs, modulating post-transcriptional gene expression and influencing key processes such as apoptosis, autophagy, oxidative stress, inflammation, and tissue remodeling.

Certain circRNAs have been identified as pro-damaging agents that promote cardiomyocyte loss and pathological cardiac remodeling. For example, circNCX1 (also known as circSLc8a1)**,** derived from sodium-calcium exchanger gene *NCX1*, promotes cardiomyocyte apoptosis during ischemia through suppression of miR-133a-3p activity and targeting CDIP1 [171]. circNfix inhibits cardiac regeneration and tissue recovery by inducing Ybx1 ubiquitin-mediated degradation and targeting Gsk3β via miR-214 sponging [204]. Similarly, CDR1as, primarily expressed in the brain [66], accelerate post-infarction dysfunction by sequestering miR-7a and promoting pro-apoptotic gene expression in cardiomyocytes, thus resulting in compromised post-infarction function in mouse models [201]. MFACR also induces cell death through regulation of miR-125b gene methylation [197] and promotes mitochondrial fission via the miR-652-3p/MTP18 pathway [208]. Downregulation of circZNF609 has been reported to correlate with improved cardiac repair through epigenetic modulation of YAP expression [195], whereas circHIPK2 induces apoptosis and maladaptive autophagy through miR-485-5p-mediated regulation of ATG101 after oxidative injury [192]. Isoforms of circHIPK3 [42,189] and circSAMD4A [190] have also been reported to increase apoptosis, oxidative stress, and inflammation.

Conversely, other circRNAs play a protective function, reducing damage and promoting tissue repair. For instance, ACR inhibits excessive activation of autophagy during I/R injury by regulating the Pink1/FAM65B pathway [198], while circPAN3 functions similarly by targeting the miR-421/Pink1 pathway [191], circFOXO3 reduces I/R injury by inhibiting HMGB1/KAT7-induced autophagy [206], and circFNDC3B assists in post-infarction repair by reducing cardiomyocyte and endothelial cell death and improving angiogenesis through the FUS/VEGF-A axis [222]. CNEACR promotes cell survival by negatively regulating cardiomyocyte necroptosis induced by histone deacetylase (HDAC) activity [223]. Finally, some circHIPK3 and circSAMD4 isoforms cause improvement of post-infarction cardiac function through the modulation of signaling pathways such as oxidative stress [221], growth and angiogenesis [212,216], and mitochondrial biogenesis and function [207]. These circRNAs present a biphasic nature capable of both promoting damage and improving cardiac regeneration and neovascularization, depending on the context.

Overall, circRNA expression patterns after infarction and reperfusion reflect a complex and counter-balanced network, in which some circRNAs promote death and pathological remodeling mechanisms, whereas others favor cell survival, angiogenesis, and tissue repair. This dual functionality positions circRNAs as promising prognostic biomarkers and potential therapeutic targets for modulating myocardial response to ischemic injury.

### 5.2. Cardiac Hypertrophy (CH), Cardiac Fibrosis (CF), and Heart Failure (HF)

Cardiac hypertrophy and fibrosis represent pathological adaptations to hemodynamic overload and neurohormonal stress that ultimately culminate in heart failure. Progression to heart failure involves myocardial structural and functional alterations in which circRNAs are emerging as important regulators of cell survival, stress responses, fibroblast proliferation, and tissue remodeling [186].

Within this context, some circRNAs demonstrate protective functions, limiting maladaptive remodeling processes, while others act as pro-pathological factors, inducing cardiac hypertrophy, fibrogenesis, and functional loss. Among the protective circRNAs, HRCR has also been identified to be an important suppressor of pathological hypertrophy by sponging miR-223, preventing dysregulation of downstream targets and limiting cardiac decompensation [202]. Similarly, circNfix diminishes pressure overload and induces hypertrophy via the miR-145-5p/ATF3 pathway [203], while elevated circNFIB levels antagonize cardiac fibrosis by inhibiting fibroblast proliferation by sponging miR-433 [213]. Additionally, ACR, previously described in relation to I/R injury, correlates with improved prognosis in chronic heart failure, suggesting a long-term cytoprotective role [193].

Conversely, several circRNAs promote hypertrophy, fibrogenesis, and functional decline. circNCX1/circSlc8a1 is highly expressed in cardiomyocytes, and its knockdown mitigates pressure-overload-induced hypertrophy [205]. circHIPK3 has been extensively studied in this context: its expression causes hypertrophy through the miR-185-3p/CASR pathway [194], worsens heart failure by increasing adrenergic signaling via miR-17-3p/ADCY6 activation [214], and raises fibroblast proliferation, migration, and phenotype switching, leading to cardiac fibrosis, both via angiotensin II-induced sequestration of miR-29b-3p [215] and via the miR-152-3p/TGF-β2 pathway under hypoxic conditions [217]. Moreover, circCDR1as, which has been previously associated with myocardial infarction, is also involved in cardiomyocyte proliferation and apoptosis regulation via the miR-135a/b-HMOX1 axes, contributing to contractile dysfunction and chronic heart failure post-infarction [199,230]. Ultimately, circRNA_000203 directly mediates hypertrophy by suppressing miR-26b-5p and miR-140-3p, causing GATA4 dysregulation, a critical transcriptional factor of cardiac growth [219].

Overall, these findings demonstrate circRNAs are central hubs orchestrating the switch from adaptive to pathological cardiac remodeling. Some, such as HRCR, circNfix, circNFIB, and ACR, slow the process, controlling hypertrophy and fibrosis and preserving cardiac function. Others, such as circHIPK3, circCDR1as, and circRNA_000203, are stimulators of maladaptive molecular circuits, contributing to the development of hypertrophic cardiomyopathy and heart failure. Understanding these dynamics implicates circRNAs as potential candidates both as prognostic biomarkers and as therapeutic targets for early intervention of cardiac remodeling.

### 5.3. Vascular Pathologies: Atherosclerosis (AS) and Vascular Calcification (VC)

Vascular diseases such as atherosclerosis and vascular calcification significantly contribute to cardiovascular morbidity by promoting myocardial ischemia, organ failure, and thrombotic complications. The altered balance between cell proliferation, senescence, and differentiation is partly modulated by circRNAs [236,237]. Some studies have shown that circANRIL regulates rRNA maturation by interacting with the PES1 protein, a component of the pre-rRNA maturation complex. Interaction with PES1 compromises ribosomal biogenesis, inducing apoptosis and reducing the proliferation of smooth muscle cells and vascular progenitor cells, with a consequent protective effect against atherosclerotic progression [47].

In contrast, some circRNAs act in a pro-atherogenic manner. For instance, circCHFR promotes the proliferation and migration of vascular smooth muscle cells (VSMCs) by interacting with several regulatory axes: on the one hand, by sequestering miR-370, this results in an increase in FOXO1 [210], and on the other hand, by acting on the miR-214-3p/Wnt3-β catenin pathway, it stimulates the phenotypic transition and migration of VSMCs [218].

In endothelial cells, meanwhile, certain circRNAs modulate cellular senescence. Indeed, circGNAQ and ciPVT1 exert a protective role, limiting cellular aging processes and preserving vascular integrity. In particular, circGNAQ acts as a miRNA sponge for miR-146a, an interaction that leads to an increase in the expression of anti-senescence and endothelial protective genes [227]. Whereas ciPVT1, by sequestering pro-senescence miRNAs such as miR-125b and miR-146a, maintains a “young” and anti-calcifying endothelial phenotype, reducing signals that promote the osteogenic transition of interstitial valve cells [228].

Vascular calcification (VC), a complex pathological process involving the osteogenic differentiation of vascular cells, is modulated by several circRNAs. One example is circSAMD4, which shows an inverse correlation with VC severity by sponging miR-125a-3p, which promotes osteogenic differentiation, resulting in the activation of a previously silenced anti-osteogenic transcription factor, Ets1. Upon this activation, a reduction in mineralization and osteoblastic transition in vascular and valvular interstitial cells is observed [238]. Due to its protective role against calcification, circSAMD4 is proposed as a predictive biomarker for VC and calcific aortic valve disease (CAVD) severity [224]. Conversely, circARHGAP10 acts by sponging miR-335-3p, which normally suppresses expression of RUNX2, a master regulator of osteogenesis. Upregulation of RUNX2 promotes osteogenic differentiation of valvular interstitial cells (VICs), accelerating calcium deposition in valve leaflets [226], and contributes to valve calcification and aortic stenosis progression.

### 5.4. Pulmonary Hypertension (PH)

Pulmonary hypertension (PH) is characterized by progressive vascular remodeling involving the proliferation and aberrant migration of pulmonary artery smooth muscle cells (PASMCs), endothelial dysfunction and activation, and increased extracellular matrix deposition. These changes lead to elevated pulmonary vascular resistance and ultimately right ventricular failure [239,240]. While canonical signaling pathways such as TGF-β, BMP, Notch, and Wnt are well-known mediators of cellular changes, circRNAs have recently emerged as critical regulators of these pathological processes. They function both as sponges for miRNAs and as scaffolds for regulatory proteins or templates for unconventional translation.

A prime example is CDR1as, one of the most studied circRNAs in cardiovascular diseases. In PH models, CDR1as promotes osteogenic differentiation and calcification of PASMCs by sequestrating miR-7-5p and releasing its targets CNN3 (calponin 3) and CAMK2D (Ca^2+^/calmodulin-dependent protein kinase II delta) from inhibition. The activation of this miR-7-5p/CNN3-CAMK2D axis promotes cytoskeletal reorganization, contractile activity, and Ca^2+^-dependent signaling, enabling PASMCs to acquire an osteogenic and proliferative phenotype that contributes to vascular remodeling and arterial wall stiffness. This effect positions CDR1as as a molecular driver of the dysfunctional vascular calcification characteristic of PH [225].

At the endothelial level, circ5078 modulates pulmonary endothelial cell proliferation by regulating the availability of Caprin-1, a key protein for stress granule (SG) formation. Stress granules are cytoplasmic complexes responsible for sequestering mRNA under cellular stress, limiting global protein translation, and promoting cell survival. The reduction in SG formation mediated by circ5078 results in increased translation of mitoribosomal proteins, increasing mitochondrial metabolism and the proliferative capacity of endothelial cells. Hence, circ5078 contributes to endothelial hyperplasia and vascular homeostasis disruption, two hallmark events in early pulmonary vascular remodeling [220].

### 5.5. Diabetes Mellitus (DM) and Diabetic Cardiomyopathy (DCM)

Diabetes mellitus is a major risk factor for cardiovascular disease, and its cardiac complication, diabetic cardiomyopathy (DCM), are significant cardiovascular risk factors characterized by cardiomyocyte hypertrophy and apoptosis, interstitial fibrosis, endothelial dysfunction, and alterations in energy metabolism. These events are strongly driven by chronic hyperglycemia and oxidative stress, which promote redox imbalances, activation of pro-apoptotic kinases, and pathological remodeling of the myocardium [241,242]. In recent years, it has emerged that circRNAs play a crucial role in modulating these processes through interactions with miRNAs and key signaling pathways.

One of the main pro-apoptotic regulators is CDR1as. In mouse models of DCM, CDR1as acts as a molecular sponge capable of activating the Hippo pathway, promoting the activation of the MST1/LATS2-YAP cascade. The activation of MST1 (mammalian sterile 20-like kinase 1) causes phosphorylation and inhibition of YAP, a transcription factor normally involved in cell survival and growth. The dysregulation of this axis promotes cardiomyocyte apoptosis and loss of contractile mass, aggravating ventricular dysfunction [200].

Concurrently, circMAP3K5 was found to be closely associated with the progression of DCM. MAP3K5 (also known as ASK1, apoptosis signal-regulating kinase 1) is a kinase activated by oxidative stress that acts as a central junction in the pro-apoptotic JNK/p38 pathways. circMAP3K5, overexpressed in diabetic hearts, acts by sequestering miR-22-3p, a miRNA with a cytoprotective function. The inhibition of miR-22-3p causes an increase in the expression of DAPK2 (death-associated protein kinase 2), a pro-apoptotic serine/threonine kinase that amplifies the cell death cascade. In this way, circMAP3K5 enhances the activation of stress-related MAPK pathways, contributing to cardiomyocyte loss and contractile dysfunction [196].

In contrast to CDR1as and circMAP3K5, which have predominantly pro-apoptotic effects, circHIPK3 shows a more heterogeneous action profile linked to vascular dysfunction in diabetes. Under conditions of hyperglycemia, circHIPK3 derived from endothelial cells is enriched in exosomes and transferred to vascular smooth muscle cells (VSMCs). There it acts as a sponge for miR-106a-5p, releasing its targets Foxo1 and Vcam1, which promote cell survival and inflammatory activation of the endothelium, respectively. This mechanism stimulates abnormal VSMC proliferation and endothelial activation, promoting vascular thickening, stiffness, and altered flow homeostasis, all distinctive features of diabetic vasculopathy [211].

Notably, circRNAs exhibit cell-functional versatility that can be context-dependent, manifesting either protective or deleterious effects in cardiovascular tissues [243,244]. In fact, their role is not restricted to a single function but extends across several. These molecules act as central regulators within the cell, with effects that may shift based on the pathological setting, sometimes promoting cell survival and tissue repair, and at other time worsening disease-associated damage [245]. This dual effect is influenced by their molecular interaction partners, which may vary according to their pathological states (e.g., circFOXO3 [174,206,229], circHIPK3 [42,189,194,214,215,216,217]). Moreover, due to the complexity of their function, circRNAs often have distinct, and sometimes opposite roles compared to their linear counterparts (e.g., circFOXO3 [174,229] vs. FOXO3 [246,247] and lncANRIL [248] vs. circANRIL [47]).

A significant aspect of circRNA biology in the cardiovascular field is their association with extracellular vesicles (EVs), particularly exosomes, which mediate intercellular communication under both physiological and pathological conditions (e.g., circHIPK3 [211,212,221]). EVs can transport circRNAs systemically through the bloodstream, potentially modulating distant target cells. This mechanism presents promising applications for circRNA-based diagnostics, therapeutic delivery, and biomarker discovery, which have been extensively explored in oncology but remain under-investigated in CVDs. Despite the growing understanding of circRNAs in cardiovascular pathology, translation of these findings into clinical applications is still in its early stages.

Given the fundamental involvement of circRNAs as a key factor in the pathophysiology of CVDs, further comprehensive studies are essential to fully elucidate their diagnostic, prognostic, and therapeutic potential, facilitating translation from bench to bedside.

## 6. Clinical Applications

In recent years, medicine worldwide has increasingly focused on a patient-specific approach. Diagnostic and prognostic molecular biomarkers enable more precise patient stratification and guide targeted therapies based on individual molecular profiles. While this paradigm shift is most prominent in oncology, it is progressively extending to other major disease categories, including CVDs. The clinical potential of RNA-based technologies was first prominently demonstrated using mRNA vaccines against COVID-19 infections [249]. Since then, interest in using the transcriptome for diagnostic, prognostic, and therapeutic purposes has markedly expanded. In the context of circRNAs, clinical applications remain in the early developmental stages; however, the considerable promise of these molecules as novel biomarkers and therapeutic agents is increasingly recognized.

The clinicalTrials.gov database reveals a limited number of ongoing clinical trials investigating circRNAs as biomarkers or therapeutic agents in CVDs. Current research predominantly focuses on quantifying and characterizing circRNA expression profiles, either individually or in combination with other non-coding RNAs (ncRNAs), such as miRNAs and long non-coding RNAs, to evaluate their potential as diagnostic or prognostic biomarkers. Most biomarker studies emphasize expression profiling of individual or combined circRNAs alongside other ncRNAs (e.g., NCT03170830, NCT05098340, NCT04230785, NCT0417569, NCT02297776).

Therapeutic strategies under investigation include the administration of synthetic circRNAs, such as HM2002, currently being evaluated for safety and tolerability in patients with ischemic heart failure (NCT06621576). The targeted clinical conditions include stroke, cardiac arrest, myocardial infarction, and heart failure, with delivery methods ranging from systemic administration to localized epicardial injection.

### 6.1. Useful Molecular and Functional Features of Circular RNAs

CircRNAs have distinct molecular properties that make them promising candidates for translational applications. Their covalently closed circular structure confers resistance to exonuclease-mediated degradation and therefore exhibits greater stability than their linear mRNA counterparts [6,250]. While most linear mRNAs have a lifespan of only a few hours, circRNAs typically persist for much longer, with a median half-life exceeding ten hours [6,250]. This enhanced stability is particularly advantageous in therapeutic settings, as it facilitates a long-lasting cellular effect, potentially reducing the frequency and toxicity of repeated administrations [251].

Another feature that makes circRNAs potentially valuable for clinical applications is their often high cell- and tissue-specific expression and their role as central regulators of cellular function through extensive interaction with various biomolecules. Not only do they participate in intracellular processes involving transcription, translation, and post-translational modifications, but they also engage in intercellular communication through their enrichment in extracellular vesicles such as exosomes [252,253]. These vesicles enable circRNAs to be transported systemically, influencing distant cells and contributing to complex signaling networks. The immunogenicity of circRNAs remains a debated topic [254]. While early studies suggested that exogenous circRNA administration could activate immune responses through the RIG-I pathway [173], subsequent research has indicated that these effects might derive from contaminants associated with in vitro transcription [255]. Endogenous circRNAs mitigate involuntary immune activation via methylation modifications [256], and in vitro transcribed circRNAs may bind and sequester the protein kinase PKR, modulating its activity [257]. Overall, immune activation appears independent of intrinsic circRNA features.

### 6.2. Diagnostic Potential: Circulating Biomarkers

Non-coding RNAs, particularly miRNAs and long non-coding RNAs (lncRNAs), have been extensively considered as diagnostic and prognostic biomarkers in multifactorial diseases such as cardiovascular conditions [258]. The current treatments focus primarily on risk-factor control, early diagnosis, and customized pharmaceutical therapy. However, the available approaches are still not very satisfactory. Traditional biomarkers (cTnT, CK-MB, myoglobin, BNP, and CRP) provide a fundamental aid by reflecting myocardial function, heart function, and inflammatory status, but they have limitations in terms of sensitivity and specificity. This sparks interest in new biomarkers, such as circRNAs, which offer promising characteristics: remarkable stability in both intracellular and extracellular fluids, cell- and tissue-specific expression, and presence in blood plasma and serum. Their enrichment in extracellular vesicles facilitates paracrine and endocrine signaling, allowing circRNAs to influence recipient cells at distant sites [259]. In CVDs, the analysis of diseased tissues at the source is very complicated and often limited, making liquid biopsies, such as blood sampling, the primary non-invasive approach for obtaining patient specimens [260]. In this context, thanks to their stability and presence in body fluids, circRNAs can be effectively utilized as circulating biomarkers. One well-studied biomarker example is MICRA (Myocardial Infarction-associated circRNA), proposed as a prognostic biomarker following myocardial infarction. Studies have demonstrated that MICRA levels are significantly reduced in the blood of post-infarction patients, and its low abundance correlates with the risk of left ventricular dysfunction several months after the event [261]. Furthermore, the prognostic value of MICRA may be enhanced when combined with established clinical variables such as NT-proBNP and creatine phosphokinase (CPK), improving risk stratification for patient outcomes [262]. Beyond MICRA, multiple circRNAs have been identified in circulation, reflecting their stable presence and potential utility across several cardiovascular conditions [263,264,265,266,267].

Despite the promising characteristics of circRNAs and numerous preliminary findings in the CVD field, there is insufficient consensus within the scientific community regarding their reliability and therapeutic efficacy [268,269]. New discovery studies should be rigorously designed to generate robust data, using larger, well-characterized cohorts that more accurately represent population diversity, including proper controls and comprehensive patient information such as treatment regimens, comorbidities, and diagnostic procedures. Current research predominantly aims at identifying individual circRNAs as disease biomarkers; however, an alternative strategy involves characterizing a specific signature of circRNAs, potentially integrating with clinical parameters and established circulating biomarkers [89]. This integrative approach could improve the robustness of the data, supported by artificial intelligence-driven analytics.

### 6.3. CircRNA-Based Therapies

The development of circRNA-based therapies is currently in the preclinical phase due to several unresolved technical and biological challenges [270,271]. Once these limitations are overcome, both endogenous and engineered circRNAs have the potential to advance into clinical applications, surpassing other RNA classes in therapeutic efficacy and stability [251]. Therapeutic strategies involving circRNAs are directly linked to their different functional mechanisms.

#### 6.3.1. miRNA Interaction

MiRNA sponging is the most widely hypothesized and characterized function of circRNAs, especially in cardiovascular research. CircRNAs can sequester pathogenic miRNAs without inducing RNAi-mediated RNA degradation, modulating miRNA activity. Moreover, engineered circRNAs containing specific miRNA binding sites have been developed to target disease-associated miRNAs [272]. Artificial circRNA sponges (cicmiRs) designed to bind hypertrophic miRNAs such as miR-132 and miR-212 [273] have demonstrated superior stability and efficacy compared to conventional linear miRNA inhibitors (antagomiRs), both in vitro and in vivo. This approach reduces cardiac hypertrophy, decreases disease progression, and restores cardiac function in transversal aortic constriction (TAC)mouse models [274].

#### 6.3.2. RNA Interaction

CircRNAs can interact directly with both pre-mRNA and mature mRNA, influencing processes such as alternative splicing and translational inhibition, respectively. For instance, circRNA-mediated exon skipping can reduce the pathogenic effects of specific exons without altering the genomic DNA [251]. This strategy has been successfully applied to restore dystrophin expression in Duchenne muscular dystrophy [275]. Extensive sequence complementarity between circRNAs and mature mRNAs can inhibit its translation [276]. Furthermore, circRNAs can be engineered to target RNA secondary structures that sequester pathogenic proteins, regulating RNA function via sequence-specific binding [251].

#### 6.3.3. Protein Interaction

Interactions between circRNAs and RBPs can either sequester pathogenic RBPs, mitigating their harmful effects, or serve as scaffolds facilitating protein-complex assembly. CircRNAs can also act as templates for protein synthesis and mediate targeted protein degradation via proteolysis-targeting chimeras complex (PROTACs), whereby circRNA-bound proteins are ubiquitinated and degraded through the proteasome pathway [277]. This application has therapeutic potential in diseases characterized by pathogenic protein accumulation, such as prion diseases [278].

#### 6.3.4. Modulation of CircRNA Expression

Therapeutic approaches include either silencing harmful endogenous circRNAs or delivering exogenous circRNAs. When silencing circRNAs the subcellular localization of the target must be carefully considered to ensure that therapeutic molecules effectively reach their site of action. Knowledge of the BSJ sequence enables specific targeting of circRNAs without affecting linear transcripts. RNA interference using siRNAs or the CRISPR/Cas13 system can suppress circRNA expression [279]; however, CRISPR/Cas13 requires further characterization regarding off-target effects. Conversely, exogenous circRNAs can be introduced by direct administration of in vitro-transcribed circRNAs or via transient expression through plasmid transfection [251]. 

#### 6.3.5. Genome Editing

Taking advantage of their stability and ability to interact with both nucleic acids and proteins, circRNAs have been proposed to support nucleic acid editing [280]. Circular guide RNAs used by the CRISPR/Cas9 system allow more efficient DNA editing due to their greater stability [281]. In prime editing, several targets can be encoded within the same circRNA, allowing simultaneous editing of several genes [280]. The interaction ability of circRNAs can also be exploited to facilitate RNA editing, by recruiting modifying proteins to the target RNA (acting as a scaffold) [282]. Overall, circRNAs exhibit improved stability and efficiency compared to their linear RNA counterparts in these contexts as well [279].

#### 6.3.6. Protein Production

Translation of circRNAs into proteins represents a promising therapeutic opportunity. The high stability of these molecules and their ability to undergo cap-independent translation are favorable features for the use in targeted therapies. CircRNA-based vaccines against infectious diseases [283,284], cancer [285], and chimeric antigen receptor T cells (CAR-T) [286] have shown potential due to prolonged antigen expression. In gene therapy, circRNA expression via viral vectors could reduce dosing frequency compared to mRNA or protein replacement therapies [251].

## 7. Current Challenges in Clinical Translation in CVDs

### 7.1. Challenges of Clinical Applications

Several circRNAs have been implicated in both the physiological and pathological processes of CVDs, and this list continues to expand. However, functional characterization and validation remain crucial to elucidating circRNA activity within target cells. Designing therapeutic circRNAs necessitates careful optimization of molecular stability, immunogenicity, and selection of delivery vehicles to ensure specific uptake by target cells.

Among primary challenges, achieving targeted inhibition of endogenous circRNAs without perturbing their linear mRNAs and delivering exogenous circRNAs with high specificity and minimal off-target effects, are the most relevant. Administration of IVT-circRNAs carries potential immunogenic risks if purification from co-products and impurities is insufficient. Conversely, vector-mediated circRNA expression, often employing viral vectors, offers longer-term stability but faces difficulties related to biogenesis efficiency and precise delivery. CircRNA production frequently depends on repeated and inverted sequence elements that facilitate back-splicing, mimicking endogenous mechanisms. However, undesired unspecific products or artifacts can be generated during synthesis, potentially compromising circRNA functionality [251]. Consequently, sequencing analyses of expression vectors are indispensable to ensure that exogenous circRNA production faithfully recapitulates the physiological cellular functions, minimizing linear RNA contaminants. Viral vectors such as adeno-associated viruses (AAV) and lentiviruses provide efficient cell- and tissue-specific delivery and appropriate RNA protection, allowing prolonged therapeutic effects for both coding and non-coding circRNA functions [287]. Nevertheless, immune responses caused by repeated administration still obstruct their clinical use [288]. Alternative delivery platforms, including lipid nanoparticles (LNPs) commonly used in mRNA vaccines, offer consistent RNA stabilization but, on the other hand, present limited tissue specificity and predominant accumulation in the liver with possible undesired effects [289]. Emerging delivery modalities, such as macrovesicles or extracellular vesicles, represent a promising approach by using endogenous circRNA transport mechanisms [290]. These physiological carriers may improve molecular stability and enable more precise targeting within the bloodstream towards specific cell types [291]. In summary, the development of circRNA-based therapy requires consideration of different aspects, including a specific vector design able to replicate physiological circRNA biogenesis while minimizing off-target RNA production and optimized delivery systems to maximize cellular uptake and avoid immunogenicity. Continued advancements in delivery technologies, particularly those based on natural carriers (extracellular vesicles) able to mimic the physiological transport mechanisms, hold great potential to overcome current limitations and advance clinical translation, especially in the cardiovascular field.

### 7.2. Challenges in Cardiovascular Research

The cardiovascular system (CVS) has a noteworthy tissue-level complexity: the heart is composed of multiple cell types (fibroblasts, endothelial cells, immune cells, and conductor cells), which create a dynamic and heterogeneous microenvironment [292,293]. This unique organization makes experimental research challenging because available models, whether in vitro or in vivo, only partially replicate cellular interactions and physiological conditions [294,295]. Nowadays, there is still a lack of a comprehensive picture of the pathophysiology of circRNAs in cardiovascular diseases due to poor research interest in certain pathologies, such as hypertension and arrhythmias different from atrial fibrillation [187].

Further criticism of the biological and technical aspects of the study of circRNAs in CVS relates to the unique characteristics of these non-coding RNAs. Adult cardiomyocytes have a low turnover rate (<1%–2%/year) with few cellular divisions; therefore, changes in circRNAs related to proliferation are limited in normal conditions. Conversely, in neonatal hearts, where proliferation is active, circRNAs associated with the cellular cycle are much more prevalent (e.g., circHIPK3 [216]) [296]. In the adult heart, acute events, such as infarction, ischemia, and inflammation, can trigger a transitory and oscillatory response that can effectively change splicing, and as a result, the biogenesis of circRNAs, making their expression dynamic. These changes are probably insufficient to make significant changes to expression profiles, so a consistent number of samples is necessary for expression changes to be detectable [297].

Sampling timing is another crucial factor; the transitory oscillation of circRNA expression, induced by pathophysiological changes, has a specific temporal range in which it is possible to be detected. It is therefore necessary to evaluate the correct timing for biological sample withdrawal [297,298]. For this reason, transversal studies, which are not longitudinal, do not allow for the monitoring and analysis of the temporal dynamics of the variation in circRNA expression in CVDs [299]. A further challenge is the scarcity of longitudinal clinical studies specifically focused on circRNAs in cardiovascular diseases.

It should also be noted that adult cardiomyocytes are difficult to isolate because they are large, fragile cells that are rich in cytoplasmic structures like sarcomeres and mitochondria, which lead to easy degradation and changes in RNA profiles during dissociation, ultimately introducing significant bias into studies [300].

It is also worth highlighting that latest-generation techniques, such as single-cell sequencing and spatial omics, have limitations in the analysis of postmitotic cardiomyocytes because of their exclusive dimensions, despite typically only accounting for one-third of adult heart cells [301]. Innovative isolation protocols, such as laser capture microdissection (LCM), can aid in the isolation of individual cardiomyocytes, but require a high rate of work to prevent RNA degradation [300].

Another important point to note is that in RNA-seq or full-length single-cell RNA-seq, the mitochondrial transcript fraction is typically used as a filter and a quality metric. This RNA population is physiologically high in the heart because cardiomyocytes are rich in mitochondria due to their enormous energy requirement [301]. Cardiomyocytes (or other key populations) that are physically rich in mitochondria are at risk of being excluded by the cutoff standard (e.g., mtRNA >5%), which results in a minor sensitivity to detection of transcripts that are low-expressed, such as the majority of circRNAs [302].

From a prospective standpoint, the combination of new generation RNA-seq, bioinformatic analysis, and artificial intelligence will enable the creation of complete databases and the use of exosomal circRNAs (which can be easily detected in plasma/serum) as non-invasive tools for early diagnosis, prognosis, and disease monitoring. Finally, it is crucial to remember that CVDs are heterogeneous, meaning that a “one-size-fits-all” approach is unlikely due to differences in sex, severity, etiology, and therapeutic response [303]. Personal molecular characterization of the patient’s RNA profile is a key goal for precise and tailored cardiovascular medicine [304].

## 8. Future Perspectives

Despite the challenges, the clinical translation of circRNAs is promising. CircRNAs possess unique features that position them as promising candidates for future translational medicine. Essential steps include standardizing analytical methods (from sample collection to bioinformatic data analysis) to ensure reproducibility, validating existing data, and conducting larger, rigorously designed multicenter and longitudinal clinical studies. From a therapeutic standpoint, the development of effective strategies focuses on precise molecular design and targeted delivery to maximize efficacy while minimizing off-target effects and unwanted immune responses. CircRNAs used as biomarkers still require optimization of standard operating techniques and timescales necessary for detection, as well as a reduction in costs to make it an easily accessible procedure in clinics. Ongoing research investigation will be crucial to unlocking their full therapeutic potential.

In the long term, circRNAs have potential applications beyond their role as biomarkers and therapeutic agents, extending into regenerative medicine. Their involvement in regulating cell proliferation, apoptosis, and differentiation suggests that circRNAs could act as key regulators of cardiomyocyte survival and vascular repair covering an area where there is an enormous unmet need for intervention in cardiovascular disease clinical management.

Although still at an early stage, circRNAs represent a revolutionary paradigm in cardiovascular biology. Continued investment in mechanistic research, technological innovation, and translational studies will be essential to fulfill their maximum therapeutic potential and ultimately to leverage circRNAs as clinically actionable tools in combating cardiovascular diseases.

## Figures and Tables

**Figure 1 ijms-26-09725-f001:**
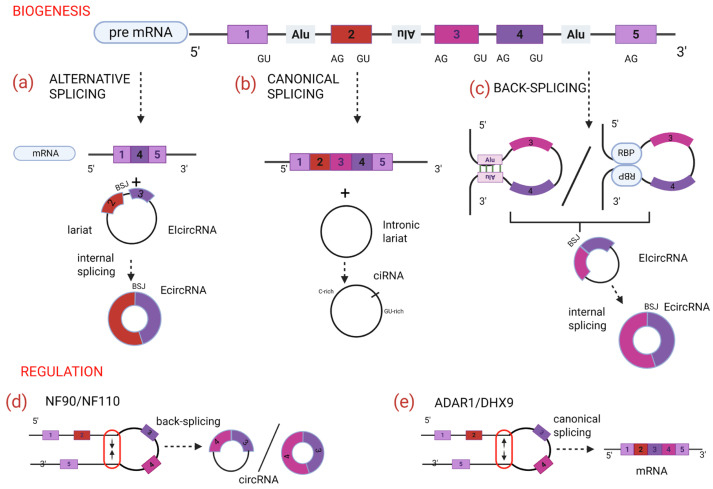
Mechanisms of circRNA biogenesis. Panel (**a**–**c**); Biogenesis Regulation. Panel (**d**,**e**). Panel (**a**) Alternative splicing: During exon skipping, the lariat containing the excised exons may undergo internal splicing. This can lead to the formation of an Exon circRNAs (EcircRNA). Panel (**b**) Canonical splicing: the excised intronic lariat can escape enzymatic degradation and form an intronic circRNA (ciRNA). Panel (**c**) Backsplicing: This mechanism can be facilitated by repeated intronic sequences and/or dimerizing RBPs, leading to the formation of circRNAs with variable composition (EIcircRNA). Panel (**d**) The formation of circRNAs can be promoted by RNA-binding factors that can stabilize circularization (NF90/NF110); panel (**e**) Other factors (ADAR1/DHX9) destabilize the duplex, promoting canonical splicing and mRNA formation. (Created with BioRender.com). ADAR1: adenosine deaminase acting on RNA; BSJ: Back-splice junction; ciRNA: intronic circRNA; circRNA: circular RNA; DHX9: ATP-dependent RNA helicase A; EIcircRNA: exon-intron circRNA; EcircRNA: exon circRNA; RBP: RNA binding protein.

**Figure 2 ijms-26-09725-f002:**
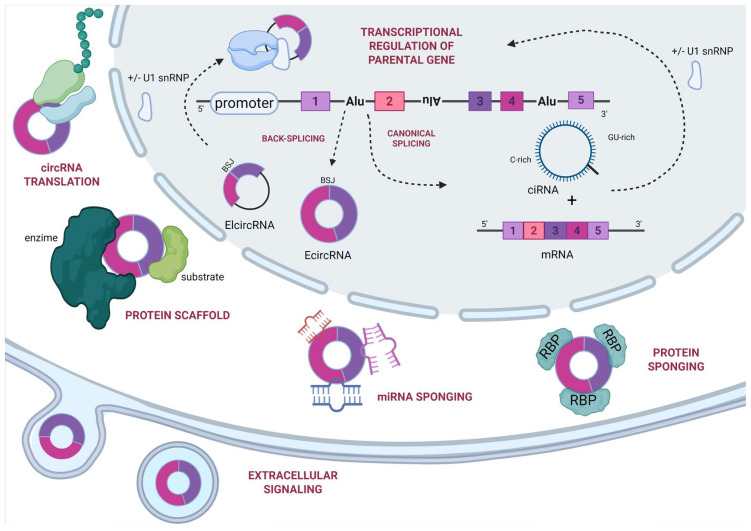
Functions of circRNAs in nucleus and cytoplasm. Nuclear circRNAs containing intronic sequences (EIcicRNAs and ciRNAs) regulate parental gene transcription by interacting with RNA polymerase II through U1 snRNP binding. Cytoplasmic exonic circRNAs (EcircRNAs) act as protein and miRNA sponges, protein scaffolds, mediate intercellular communication, and some can be translated into peptides (Created with BioRender.com). BSJ: Back-splice junction; ciRNA: intronic circRNA; circRNA: circular RNA; EIcircRNA: exon-intron circRNA; EcircRNA: exon circRNA; RBP: RNA binding protein.

**Table 2 ijms-26-09725-t002:** Bioinformatic tools for circRNA detection, full-length reconstruction, and prediction.

Tool	Application	Description	Refs
Detection			
find_circ	Split-alignment-based detection	Search for circRNAs from unmapped reads after initial genome alignment with Bowtie2	[67]
circRNA_finder	Split-alignment-based detection; Chimeric reads alignment	Uses STAR chimeric reads to detect circRNAs	[106]
CIRI	Split-alignment-based detection	Original CIRI method using paired chiastic clipping for BSJ detection	[32]
CIRI2	Split-alignment-based detection	Improved CIRI with better speed and sensitivity	[107]
DCC	Split-alignment-based detection; Chimeric reads alignment	Apply stringent statistical filters on BSJ candidates identified from STAR alignment	[108]
CircExplorer	Split-alignment-based detection	First circRNA detection tool using fusion-read mapping by TopHat-Fusion	[9]
CircExplorer2	Split-alignment-based detection; Full-length reconstruction	Improved CircExplorer in terms of accuracy, speed, and flexibility supporting multiple aligners and providing circRNA annotation	[94]
KNIFE	Pseudo-reference-based detection	Uses pre-defined BSJ references for detection	[109]
PTESFinder	Mixed approach detection	Detects post-transcriptional exon shuffling events, including circRNAs, after Bowtie alignment to a reference transcriptome and scanning for non-colinear exon junctions. Uses multiple alignment and filtering steps to reduce false positives, requiring annotation to validate splice sites	[111]
MapSplice	Splicing-aware mapper able to detect circRNAs	Detects canonical and non-canonical splicing events	[105]
Acfs	Pseudo-reference-based detection	Strength of potential splicing sites are evaluated including canonical “GU-AG” and non-canonical splicing motifs	[110]
UROBORUS	Split-alignment-based detection	Detects circRNAs by extracting unmapped reads from initial genome alignment by TopHat, splitting them into anchors, and remapping to detect BSJs by Bowtie	[112]
segemehl	Split-alignment-based detection	A short read mapper with capabilities to detect regular splicing, trans-splicing, and gene fusion events from single-end RNA-seq data	[104]
CirComPara2	Multi-tool detection	High sensitivity and accuracy in circRNA detection, giving the possibility to use up to nine different computational pipelines	[113]
circRNAwrap	Multi-tool detection; Quantification; Reconstruction	Detection and full-length characterization in a single platform	[114]
nf-core/circrna	Multi-tool detection	Integration of seven circRNA detection tools, also includes modules for differential expression analyses and circRNA-miRNA interaction prediction	[115]
Full-length Reconstruction			
CIRI-full	Full-length reconstruction	Reconstructs full-length circRNA sequences using BSJ and reverse-overlap reads	[116]
CIRI-AS	Alternative splicing event detection	Extends CIRI pipeline to detect alternative splicing events within circRNAs, enabling isoform-level annotation	[26]
CircExplorer3	Split-alignment-based detection; Full-length reconstruction	Improves CircExplorer2 annotation phase, identifying exon skipping, alternative 5′ and 3′ splice sites, and retained introns in circular transcripts	[117]
circAST	Full-length reconstruction	Reconstructs alternative splicing isoforms of circRNAs from mapped RNA-seq reads, focusing on isoform diversity	[118]
Full-length Reconstruction			
circFL-seq	Full-length reconstruction from long read sequencing	Experimental and computational workflow for profiling full-length circRNAs using Nanopore sequencing	[119]
psirc	Full-length circRNA isoform reconstruction	Reconstructs full-length circRNA isoforms using paired end reads and probabilistic modeling	[120]
CYCLeR	Full-length reconstruction	Detects and quantifies full-length circRNA isoforms by integrating BSJ detection with exon connectivity analysis	[95]
isoCirc	Full-length reconstruction from long read sequencing	Captures and sequences full-length circRNAs at isoform resolution using Nanopore sequencing	[121]
CIRI-long	Full-length reconstruction from long read sequencing	Extends CIRI pipeline for long-read sequencing, both Nanopore and PacBio data, to reconstruct and annotate full-length circRNAs	[77]
Prediction			
PredcircRNA	Prediction (Machine Learning)	Uses sequences features to distinguish circRNAs from other long non-coding RNAs by multiple kernel learning	[122]
circDeep	Prediction (Deep Learning)	Deep learning framework for classifying short and long non-coding RNAs and circRNAs	[123]
DeepCirCode	Prediction (Deep Learning)	Utilizes convolutional neural network with nucleotide sequence as input to predict BSJs	[124]
JEDI	Prediction (Deep Learning)	Predicts circRNAs and interprets relationships among splice sites to discover BSJ hotspots within a gene region.	[125]
circCNNs	Prediction (Deep Learning)	Convolutional neural networks trained to classify circRNAs and predict functional categories	[126]
CIRI-deep	Prediction (Deep Learning)	Deep learning extension of CIRI framework for predicting circRNAs and their properties even from single-cell and spatial transcriptomics data	[127]

## Data Availability

No new data were created or analyzed in this study. Data sharing is not applicable to this article.

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
