# Peer review of "Circular RNAs in Cardiovascular Physiopathology: From Molecular Mechanisms to Therapeutic Opportunities"

_ijms, 2025, doi:10.3390/ijms26199725_

Round 1
Reviewer 1 Report
Comments and Suggestions for Authors
In this review, the authors provide a comprehensive and updated overview of circRNAs, focusing on their biogenesis, classification, biological functions, detection methodologies, and translational roles in cardiovascular physiopathology. The structure is logical, and the integration of experimental evidence and bioinformatic tools enhances the rigor of the review. The methods and content framework for synthesizing current knowledge on circRNAs in CVDs are generally acceptable. However, there are some essential problems should be addressed by authors, which are listed below.
- Gene names should be in italics, such as “ANKRD52 gene”.
- In Section 2.2, the classification of circRNAs is insufficiently detailed and described with inadequate clarity. For instance, based on the chromosomal locations from which they are derived, circRNAs can be categorized into exonic circRNAs, circular intronic RNAs, exon-intron circRNAs, intergenic circRNAs, intergene-exon circRNAs, and intergene-intron circRNAs.
- It is recommended to supplement the research status and challenges of circRNA overexpression, silencing, and knockout technologies, so as to improve the discussion on the methodologies for circRNA functional verification and mechanism research.
Author Response
We sincerely thank you for the careful evaluation of our manuscript and the constructive comments
provided. We appreciate the opportunity to improve the quality and clarity of our review. Below, we
present a detailed point-by-point response to each reviewer’s comments, describing the changes made
in the revised manuscript following your suggestions:
Comment 1
Gene names should be in italics, such as “ANKRD52 gene”.
Response:
As suggested, and in accordance with standard formatting conventions, we have carefully revised the
manuscript to ensure that all gene names are now written in italics throughout the whole text.
Comment 2
In Section 2.2, the classification of circRNAs is insufficiently detailed and described with inadequate
clarity. For instance, based on the chromosomal locations from which they are derived, circRNAs can
be categorized into exonic circRNAs, circular intronic RNAs, exon-intron circRNAs, intergenic circRNAs,
intergene-exon circRNAs, and intergene-intron circRNAs.
Response:
In accordance with your suggestion for a clearer and more comprehensive classification of circRNAs,
we have substantially expanded Section 2.2 to better reflect the current classification framework (see
page 4-5). Specifically, we now describe circRNAs according to both sequence composition (exonic,
intronic, and exon-intron) and chromosomal origin (gene-based, intergenic, read-through, and fusion).
By reorganizing and implementing this section, we believe we now provide a clearer and more
comprehensive overview of circRNA classification, in line with the reviewer’s suggestion.
Comment 3
It is recommended to supplement the research status and challenges of circRNA overexpression,
silencing, and knockout technologies, so as to improve the discussion on the methodologies for
circRNA functional verification and mechanism research.
Response:
In the revised manuscript, we have substantially expanded the section on methodological approaches
for circRNA functional studies (Section 4.5, page 16-17). As suggested, we now provide details on
current techniques for circRNA overexpression, silencing, and knockout. We also highlight their
operating principles, main challenges and limitations, as well as recommendations to ensure specificity
and reliable functional verification, to improve the content of the discussion section.
Reviewer 2 Report
Comments and Suggestions for Authors
This manuscript covers several important aspects of the background information on circRNAs. Text and Figures are well-presented and easy to understand. Authors may consider addressing the following.
- Most of the text appears dedicated to developing it as a general article on circRNAs. Nevertheless, ‘Molecular mechanisms and therapeutic potential of Circular RNAs in Cardiovascular Pathophysiology’ appears in the title of this article and it lacks a definitive focus on the importance of circRNA in normal or pathophysiological aspects of the cardiovascular system (CVS). It is understandable that this area of research is growing, and definitive information will emerge in due course but the manuscript in this iteration does not provide consolidated synthesis of the available information on the topic of interest.
- Most information on circRNA and cardiovascular system appears summarized in Table 3 with majority of the material relating to in vitro cell culture models or animal models. The legend at the bottom does not describe the contents and provides a list of abbreviations which can be made more comprehensive. An enormous effort must have gone into preparing this Table and that deserves appreciation, but Table 3 is mentioned only once in the text. Table 3 covers studies on cell death and a paragraph related to cell death appears in the text (page 19), but this paragraph does not connect with the Table. More explanation and discussion on the significance of the results summarized in this Table will be helpful.
- CVD of interest are mentioned with 2 references (#170, 171) and include cardiac hypertrophy, fibrosis, heart failure, myocardial infarction, coronary artery disease, and atherosclerosis. Metabolic syndromes including diabetes, cerebrovascular (stroke, aneurysms), and pulmonary complications are also mentioned. Authors may consider summarizing the published work related to circRNAs and these conditions under a separate sub-heading or in a Tabular format.
- Various aspects of the CV system are either not included or mentioned without much information. Summarizing information on circRNA in CV physiology, development, regeneration, diagnosis, disease biomarkers and pathophysiological conditions will be a valuable addition to develop the theme of this manuscript.
- Some sub-topics are split. For example, ongoing clinical trials related to circRNAs in CVD are mentioned but Clinical Applications of circRNAs are addressed in a separate section. Combining these will help bolster the presentation on the clinical significance of circRNAs.
- Sections on ‘Current Challenges’ and ‘Future Prospects’ do not include terms like CVS/CVD nor address the specific issues and problems related to the growing field of circRNA research in cardiovascular system or CVD. Including authors’ insights and perspectives on this topic under these sections will enhance the value of this article.
Author Response
Comment 1
Most of the text appears dedicated to developing it as a general article on circRNAs. Nevertheless,
‘Molecular mechanisms and therapeutic potential of Circular RNAs in Cardiovascular Pathophysiology’
appears in the title of this article and it lacks a definitive focus on the importance of circRNA in normal
or pathophysiological aspects of the cardiovascular system (CVS). It is understandable that this area of
research is growing, and definitive information will emerge in due course but the manuscript in this
iteration does not provide consolidated synthesis of the available information on the topic of interest.
Response:
In the revised version of the manuscript, we have substantially strengthened the focus on circRNAs in
the cardiovascular context by introducing in Section 5 (pages 19-26) dedicated sub-sections that
discuss their impact in specific cardiovascular diseases, vascular pathologies, as well as metabolic
syndromes. We believe these additions provide a clearer synthesis of the available information and
better align the manuscript with the intended scope of highlighting the relevance of circRNAs in
cardiovascular pathophysiology.
Comment 2
Most information on circRNA and cardiovascular system appears summarized in Table 3 with majority
of the material relating to in vitro cell culture models or animal models. The legend at the bottom does
not describe the contents and provides a list of abbreviations which can be made more comprehensive.
An enormous effort must have gone into preparing this Table and that deserves appreciation, but Table
3 is mentioned only once in the text. Table 3 covers studies on cell death and a paragraph related to cell
death appears in the text (page 19), but this paragraph does not connect with the Table. More
explanation and discussion on the significance of the results summarized in this Table will be helpful.
Response:
Thanks for the constructive comment and for appreciating our effort regarding Table3. In the revised
version, we have expanded the legend and provided a clearer description of the table’s content in
addition to the list of abbreviations and referred to Table 3 in multiple parts of the main text. We believe
these changes better contextualize the information presented in the table.
Comment 3
CVD of interest are mentioned with 2 references (#170, 171) and include cardiac hypertrophy, fibrosis,
heart failure, myocardial infarction, coronary artery disease, and atherosclerosis. Metabolic syndromes
including diabetes, cerebrovascular (stroke, aneurysms), and pulmonary complications are also
mentioned. Authors may consider summarizing the published work related to circRNAs and these
conditions under a separate sub-heading or in a Tabular format.
Response:
We thank the reviewer for this valuable comment. We would like to note that this point has been
addressed in our two previous responses: in the revised version we have dedicated subsections in
Section 5 (pages 19-26) covering cardiovascular diseases, vascular pathologies, and metabolic
syndromes to provide a structured summary directly linked to the corresponding pathological context.
Comment 4
Various aspects of the CV system are either not included or mentioned without much information.
Summarizing information on circRNA in CV physiology, development, regeneration, diagnosis, disease
biomarkers and pathophysiological conditions will be a valuable addition to develop the theme of this
manuscript.
Response:
In line with the scope and title of the review, the main goal of this article was to focus on the circRNAs
role in CVDs pathophysiological conditions, which represents the area where translational implications
are more evident. Nevertheless, we have expanded the manuscript including discussion regarding their
potential as diagnostic and prognostic biomarkers (Section 6, pages 26-29 Clinical Applications), as
well as their role in cardiovascular regeneration (Section 8, Future Perspectives pages 31-32). We
believe these additions provide a more comprehensive view of their role while maintaining the
translational focus of the article.
Comment 5
Some sub-topics are split. For example, ongoing clinical trials related to circRNAs in CVD are mentioned
but Clinical Applications of circRNAs are addressed in a separate section. Combining these will help
bolster the presentation on the clinical significance of circRNAs.
Response:
In the revised version, as suggested, we have integrated the discussion of ongoing clinical trials related
to the Clinical Applications of circRNAs in cardiovascular pathologies into the section 6 page 27, so that
all clinical aspects are now presented together in a more coherent and comprehensive manner.
Comment 6
Sections on ‘Current Challenges’ and ‘Future Prospects’ do not include terms like CVS/CVD nor address
the specific issues and problems related to the growing field of circRNA research in cardiovascular
system or CVD. Including authors’ insights and perspectives on this topic under these sections will
enhance the value of this article.
Response:
We have enriched Section 7 pages 30-31 with a new dedicated subsection 7.2 (Challenges in
Cardiovascular Research) that addresses the specific methodological and biological limitations of
circRNA research in the cardiovascular system, including the complexity of cardiac tissue composition,
the low turnover of adult cardiomyocytes, the scarcity of longitudinal clinical studies, and technical
issues related to sequencing and sample preparation. Furthermore, in Section 8 (Future Perspectives),
we now explicitly highlight the potential of circRNAs in cardiovascular regenerative medicine,
particularly regulating cardiomyocyte survival and vascular repair, areas of major unmet clinical need.
We believe these additions strengthen the cardiovascular focus of the manuscript, opening new ideas
for further investigations.
Round 2
Reviewer 1 Report
Comments and Suggestions for Authors
Nice work of the revision.
Reviewer 2 Report
Comments and Suggestions for Authors
Authors have thoughtfully incorporated the suggestions made during the first review of their manuscript.